# Correlated Low-Rank Adaptation for ConvNets

**Wu Ran**[1]  **Weijia Zhang**[1]  **Shuyang Pang**[2]  **Qi Zhu**[1]  **Jinfan Liu**[1]  **Jingsheng Liu**[2]
**Xin Cao**[2]  **Qiang Li**[2]  **Yichao Yan**[1]  **Chao Ma**[1,*]

[1] MoE Key Lab of Artificial Intelligence, AI Institute, Shanghai Jiao Tong University
[2] CISDI Information Technology CO., LTD.

{bonjourlemonde, weijia.zhang, georgezhu, ljflnjz, yanyichao, chaoma}@sjtu.edu.cn
{shuyang.pang, jingsheng.liu, xin.a.cao, qiang.k.li}@cisdi.com.cn

## Abstract

Low-Rank Adaptation (LoRA) methods have demonstrated considerable success in achieving parameter-efficient fine-tuning (PEFT) for Transformer-based foundation models. These methods typically fine-tune individual Transformer layers using independent LoRA adaptations. However, directly applying existing LoRA techniques to convolutional networks (ConvNets) yields unsatisfactory results due to the high correlation between the stacked sequential layers of ConvNets. To overcome this challenge, we introduce a novel framework called Correlated Low-Rank Adaptation (CoLoRA), which explicitly utilizes correlated low-rank matrices to model the inter-layer dependencies among convolutional layers. Additionally, to enhance tuning efficiency, we propose a parameter-free filtering method that enlarges the receptive field of LoRA, thus minimizing interference from non-informative local regions. Comprehensive experiments conducted across various mainstream vision tasks, including image classification, semantic segmentation, and object detection, illustrate that CoLoRA significantly advances the state-of-the-art PEFT approaches. Notably, our CoLoRA achieves superior performance with only 5% of trainable parameters, surpassing full fine-tuning in the image classification task on the VTAB-1k dataset using ConvNeXt-S. Code is available at `https://github.com/VISION-SJTU/CoLoRA`.

## 1 Introduction

Transformers [1, 2] have significantly advanced the development of large-scale pre-trained foundation models [3, 4, 5] in both natural language processing (NLP) and computer vision. To fully leverage the capabilities of these pre-trained foundation models, parameter-efficient fine-tuning (PEFT) [6, 7, 8, 9, 10, 11] has emerged as a prevalent solution. Typically, low-rank adaptation (LoRA) approaches [12, 13, 14, 15, 16] have demonstrated particular promise in Transformer-based models such as LLaMA [5] and ViT [2]. By introducing independent low-rank matrices into each layer, LoRA effectively modulates the global attention maps of Transformers, facilitating efficient adaptation to downstream tasks while significantly reducing the number of trainable parameters.

Despite its success in Transformers, the application of LoRA to convolutional networks (ConvNets) remains largely unexplored. ConvNets are often the preferred architecture for various vision tasks [17, 18, 19], such as semantic segmentation and object detection, owing to their computational efficiency and robust inductive bias. However, using LoRA to adapt pre-trained ConvNets for downstream vision tasks presents considerable challenges. Unlike Transformers, which utilize inputs with global receptive fields, ConvNets are designed with sequential convolutional layers that have limited receptive fields. Directly applying independent LoRA to each convolutional layer, similar to how it

---

*Corresponding author.

39th Conference on Neural Information Processing Systems (NeurIPS 2025).

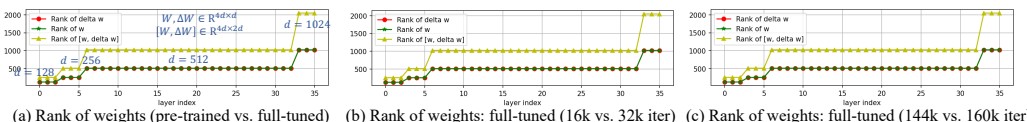

(a) Rank of weights (pre-trained vs. full-tuned)   (b) Rank of weights: full-tuned (16k vs. 32k iter)   (c) Rank of weights: full-tuned (144k vs. 160k iter)

Figure 1: Adapting a convolutional network ConvNeXt-B [17] pretrained on ImageNet to ADE20K [20] requires a full-rank tuning of the weight matrix $W$. The ranks of the weight matrix $W \in \mathbb{R}^{4d \times d}$ and the weight update matrix $\Delta W \in \mathbb{R}^{4d \times d}$ of all pwconv1 layers in ConvNeXt-B, where $d$ specifies channel dimension. We can see that both $W$ and $\Delta W$ almost remain consistent rank $d$ (full-rank) after tuning (a), at the initial (b) or final (c) stages, respectively. Moreover, when concatenating $W$ and $\Delta W$, the rank is nearly double ($2d$). As $\Delta W$ resides in a totally different parameter space from $W$, we cannot adapt ConvNets using layer-independent low-rank matrices.

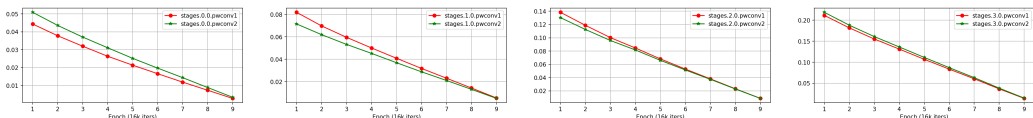

Figure 2: Fine-tuning the pre-trained ConvNeXt-B [17] on ADE20K [20]. In order to show the high correlation of adjacent convolutional layers, we compute the mean singular value of their update matrices $\Delta W_1$ (pwconv1) and $\Delta W_2$ (pwconv2) during fine-tuning. Note that their correlation becomes higher along with the fine-tuning process.

is done for Transformers, results in a high-rank weight update matrix, complicating the fine-tuning of ConveNets. To illustrate this, we compare the ranks of the weight matrix $W$ and the weight update matrix $\Delta W$ of the ConvNeXt-B [17] model pre-trained on ImageNet-22K and subsequently adapted to ADE20K. Figure 1 illustrates that the rank of the update matrix remains close to full throughout training. Fine-tuning ConvNets with such a high-rank update matrix is unlikely to be efficient. This raises a critical question: *Can we overcome the full-rank learning bottleneck in the fine-tuning of ConvNets and develop an effective LoRA method for ConvNets?*

ConvNets hierarchically stack convolutional layers to extract features, resulting in a significantly higher similarity between adjacent convolutional layers compared to Transformer layers, as highlighted by [21]. We observe that introducing independent LoRA into each convolutional layer leads to suboptimal adaptation performance, primarily because the correlations between adjacent convolutional layers are not accounted for. Therefore, we first examine the relationship between two adjacent linear layers, represented as $Y_1 = XW_1$ and $Y_2 = Y_1 W_2$. According to backpropagation, the gradients of $W_1$ and $W_2$ share a common term, $X^T \frac{\partial L}{\partial Y_2}$, which implies that their weight update matrices, $\Delta W_1$ and $\Delta W_2$, are highly correlated. For instance, as illustrated in Figure 2, we analyze the correlation between two adjacent convolution layers, namely pwconv1 and pwconv2. Their update matrices, $\Delta W_1$ and $\Delta W_2$, show a strong correlation throughout the entire tuning procedure. This correlation has also been explored using the Centered Kernel Alignment (CKA) metric in Appendix D. Thus, it is essential to consider the correlations between adjacent convolution layers when fine-tuning on downstream vision tasks.

In contrast to recent research [22, 23, 24] that focuses on tuning Transformer layers with independent LoRA, our approach explicitly considers the correlation between adjacent convolutional layers using low-rank matrices. We leverage the hierarchical architecture of ConvNets by updating these adjacent convolutional layers together through correlated low-rank matrices. Following the back-propagation rule applicable to adjacent convolution layers, we introduce an innovative weight update strategy (refer to Figure 3(c)) that explicitly binds their updates through shared low-rank matrices. This ensures that the weight update strategy effectively inherits the full-tuning behavior of the adjacent layers from a gradient perspective. To enhance the tuning efficiency of ConvNets further, we implement a filtering-based scheme that expands the receptive field of the shared low-rank matrices, thereby preventing uninformative local regions from dominating the gradients. Based on these strategies, we propose a novel and high-performance method called Correlated Low-RAnk Adaptation (CoLoRA) for ConvNets. Experimental results across the mainstream downstream image classification, semantic segmentation, and object detection tasks demonstrate that our CoLoRA significantly surpasses current state-of-the-art PEFT methods. In summary, we present the following contributions:

- We show that fine-tuning pre-trained ConvNets on downstream tasks is a full-rank learning process. Existing low-rank LoRA methods designed for Transformers cannot achieve satisfactory results for fine-tuning ConvNets.

- We propose correlated LoRA for fine-tuning ConvNets. We propose a novel weight update strategy to update adjacent convolution layers together.

- We conduct extensive validation on the classification, segmentation, and detection tasks. The proposed CoLoRA for ConvNets performs favorably against the state-of-the-art methods.

## 2 Related Work

**Parameter-efficient fine-tuning**. PEFT [25] has emerged as a practical alternative to full fine-tuning, especially for adapting large-scale pre-trained models to downstream tasks while minimizing computational overhead. Current PEFT methods can be broadly classified into three categories: prompt tuning [26], adapter-based methods [27, 27], and LoRA-based approaches [7, 28]. Prompt tuning techniques [26, 29, 30, 31] introduce learnable visual prompts for tuning pre-trained foundation models on downstream vision tasks. Among these, visual prompt tuning (VPT) is recognized as a highly parameter-efficient approach, though it may struggle with larger domain gaps or more complex downstream vision tasks, such as dense prediction. Recent adapter-based methods [27, 32, 33, 25, 34, 35] have gained traction by inserting lightweight bottleneck layers into each Transformer block, and have been extensively studied in both NLP and computer vision. For instance, the notable Mona [27] incorporates multi-scale adapters into the Swin Transformer [36] to enhance adaptation abilities to image recognition. Further research [37, 38, 39] investigates the potential of parallel networks, which adapt deep features using lightweight side modules. Meanwhile, LoRA [6] has gained popularity due to its effectiveness and efficiency. By integrating trainable low-rank matrices into each layer of Transformers, LoRA achieves performance comparable to full fine-tuning while significantly reducing the number of trainable parameters. In particular, LoRA effectively modulates the global attention maps of Transformer-based models by injecting independent low-rank matrices into each layer, which contributes to its flexibility and efficacy in tuning. Despite the considerable success of LoRA-based approaches in both NLP [7, 14] and vision Transformers [28], their applicability to ConvNets remains largely uncharted.

**LoRA and its variants**. LoRA [6] approximates the full tuning of pre-trained foundation models by employing layer-independent low-rank matrices. Recent studies have demonstrated that initialization schemes, weight update strategies, and rank values significantly influence LoRA's performance [7, 14, 10, 40, 41, 42, 22, 23, 15, 9, 43, 44]. The standard approach of Vanilla LoRA utilizes zero and random initializations for its low-rank matrices, which inherits a highly random tuning property. Meng *et al.* [7] propose that initializing LoRA with principal components derived from pre-trained weights leads to faster and more effective convergence. Concurrently, LoRA-GA [14] seeks to align LoRA weight initialization with the gradients from full tuning. In terms of weight updates, RsLoRA [40] introduces a rank-stable update strategy by reevaluating the impact of scaling factors. Subsequently, Liu *et al.* [41] break down the weight update matrix into two distinct components: magnitude and direction. Fan *et al.* [10] further enhance adaptation by integrating SVD-based initialization with a mixture of experts [45, 46] (MoE). An alternative avenue of exploration involves high-rank LoRA methods, such as SURMs [42] and HiRA [22], which aim to directly address the limitations imposed by the low-rank assumption through the introduction of higher-rank updates and more sophisticated fusion mechanisms.

Most PEFT methods have been developed within the framework of Transformer-based architectures, which benefit from their modular structure and self-attention mechanisms. However, convolutional networks [17, 18] continue to dominate many practical deployment scenarios due to their efficiency and robust inductive biases [18]. Recent attempts [47, 32] have merely adapted Transformer-based adaptation techniques for ConvNets, neglecting the unique characteristics of convolution, such as its strong correlation in hierarchical feature extraction. Furthermore, the full-rank property of weight updates in convolution layers, as demonstrated in Figure 1, poses additional challenges to the low-rank assumption that underlies most existing PEFT methods. Our work addresses this gap by explicitly modeling the correlation between adjacent convolution layers during tuning, providing a principled and efficient PEFT solution specifically designed for ConvNets.

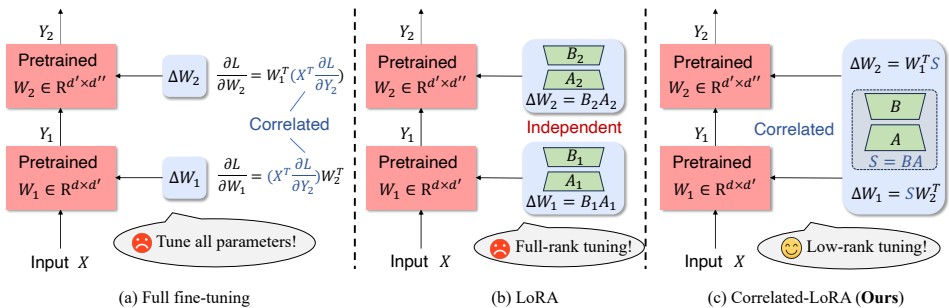

Figure 3: An intuitive comparison among full fine-tuning, existing LoRA, and the proposed CoL-oRA. (a) Full fine-tuning requires adjusting all parameters, resulting in substantial computational overhead. (b) Implementing independent full-rank LoRA within convolutional networks leads to a large number of tunable parameters. (c) In contrast, our CoLoRA facilitates low-rank tuning by capitalizing on the correlations between adjacent layers.

## 3  CoLoRA: Correlated Low-Rank Adaptation for ConvNets

### 3.1  Preliminaries

The LoRA approach [6] has emerged as a sophisticated and theoretically-grounded technique for parameter-efficient fine-tuning of large models [8, 33, 15, 48]. In essence, LoRA decomposes the update of a pre-trained weight $W_0 \in \mathbb{R}^{d \times d'}$ into low-rank matrices, as illustrated in Figure 3(b), where $d$ and $d'$ represent the input and output feature dimensions, respectively. Mathematically, the update weight with respect to $W_0$ is expressed as $\Delta W = sAB$, where $A \in \mathbb{R}^{d \times r}$, $B \in \mathbb{R}^{r \times d'}$, $r \ll \min(d, d')$ defines the rank, and $s$ serves as a scaling factor.

A useful way to understand LoRA is by comparing the gradient of $W$ in full tuning with the corresponding gradient induced by LoRA [10]. Let $\mathbf{g}_W$ denote the gradient with respect to $W$ during full tuning. We can first compute the gradients with respect to $A$ and $B$ using the following formulas: $\mathbf{g}_A = s\mathbf{g}_W B^T$ and $\mathbf{g}_B = sA^T \mathbf{g}_W$. The equivalent gradient of $W$ can then be expressed as:

$$\tilde{\mathbf{g}}_W = s\mathbf{g}_A B + sA\mathbf{g}_B = s^2(\mathbf{g}_W B^T B + AA^T \mathbf{g}_W). \tag{1}$$

Since $B$ is initialized to $\mathbf{0}$ and $A$ is randomly initialized (typically using `kaiming_uniform` [49]) with independently and identically distributed (i.i.d.) elements (mean 0 and variance $\sigma_A^2$), the equivalent gradient in Eq. (1) can be approximated as $s^2 r \sigma_A^2 \mathbf{g}_W$ (for further details, see [10]). By selecting $s_{\text{expect}} = \frac{1}{\sqrt{r}\sigma_A}$, LoRA can effectively match the full tuning scenario at the outset. However, we contend that there is a dilemma in choosing a single value of $s$ that can simultaneously align with both the expectation and variance of $\mathbf{g}_W$.

**Lemma 3.1** *If $A \in \mathbb{R}^{d \times r}$ is initialized using the Kaiming uniform method with variance $\sigma_A$, then we have $\mathrm{Var}[AA^T] \approx r\sigma_A^4 \mathbf{1}_d$, where $\mathbf{1}_d$ denotes a $d \times d$ matrix with all elements equal to 1.*

The proof is included in Appendix A. Assuming that all elements in $\mathbf{g}_W$ are i.i.d. and share a distribution of $\mathcal{N}(0, \sigma^2)$, we can calculate the variance of the elements in $\tilde{\mathbf{g}}_W$ as $\mathrm{Var}[\tilde{g}_W]_{ij} \approx s^4 r d\sigma_A^4 \sigma^2$. Consequently, using a scaling factor $s = \frac{1}{\sqrt{r}\sigma_A}$ will amplify the variance by a factor of $\frac{d}{r}$, leading to unstable tuning when the rank is small. A suitable variance matching scaling factor is derived as $s_{\text{var}} = \frac{1}{\sqrt[4]{rd}\sigma_A}$.

### 3.2  Decomposing Weight Update in Adjacent Layers

Let us begin by examining two adjacent fully connected layers, which can be viewed as a straight-forward pair of adjacent $1 \times 1$ convolutional layers. The relationships are defined as follows:

$$\begin{aligned} Y_1 &= XW_1 + \mathbf{b}_1, \\ Y_2 &= Y_1 W_2 + \mathbf{b}_2, \end{aligned} \tag{2}$$

where $X \in \mathbb{R}^{N \times d}$ represents the input with a batch size of $N$ and a dimensionality of $d$. The weight matrices are defined as $W_1 \in \mathbb{R}^{d \times d'}$ and $W_2 \in \mathbb{R}^{d' \times d''}$, while $\mathbf{b}_1$ and $\mathbf{b}_2$ denote the corresponding bias terms.

Existing LoRA-based approaches [6, 7, 10] operate under the assumption that the weight matrix $W$ can be updated using low-rank matrices, as illustrated in Figure 3(b). When incorporating LoRA updates into Eq. (2), we arrive at *independent* updates: $\Delta W_1 = s_1 A_1 B_1$ and $\Delta W_2 = s_2 A_2 B_2$. However, it is important to note that the updates $\Delta W_1$ and $\Delta W_2$ are inherently *correlated* during back-propagation. This correlation implies that stacking independent LoRA layers struggles to replicate the behavior of full training and may even undermine the hierarchical feature extraction capabilities of ConvNets. We examine this correlation from the perspective of gradients. In particular, the backward propagation rule applied to Eq. (2) yields the following relationships:

$$\frac{\partial \mathcal{L}}{\partial W_2} = Y_1^T \left( \frac{\partial L}{\partial Y_2} \right) = W_1^T \underbrace{X^T \left( \frac{\partial L}{\partial Y_2} \right)}_{correlated} + \underbrace{[\mathbf{b}_1, \cdots, \mathbf{b}_1]_N \left( \frac{\partial L}{\partial Y_2} \right)}_{independent}, \tag{3}$$

$$\frac{\partial \mathcal{L}}{\partial W_1} = X^T \left( \frac{\partial L}{\partial Y_1} \right) = \overbrace{X^T \left( \frac{\partial L}{\partial Y_2} \right)} W_2^T. \tag{4}$$

The above equations illustrate that the weight update process in consecutive layers can be decomposed into two components: a correlated component (the blue part in Eqs. (3) and (4)) and an independent component. This motivates us to represent the weight updates in adjacent layers in a general form:

$$\Delta W_1 = S W_2^T + P_1, \ \Delta W_2 = W_1^T S + P_2, \tag{5}$$

where $S$ is shared between layers, $P_1$ and $P_2$ are specific updates for each layer.

### 3.3 Correlated Low-Rank Adaptation

Similar to LoRA, the quantities $S$, $P_1$, and $P_2$ in Eq. (5) can be decomposed into low-rank matrices, leading to the proposed CoLoRA formulation:

$$\Delta W_1 = s A_s B_s W_2^T + s_1 A_1 B_1, \ \Delta W_2 = s W_1^T A_s B_s + s_2 A_2 B_2, \tag{6}$$

In this formulation, $s_1$ and $s_2$ are layer-specific scaling factors, while $s$ is shared across layers. The matrices $A_s \in \mathbb{R}^{d \times r_s}$ and $B_s \in \mathbb{R}^{r_s \times d''}$ are shared low-rank matrices with a rank of $r_s$. Conversely, $A_1 \in \mathbb{R}^{d \times r_l}$, $B_1 \in \mathbb{R}^{r_l \times d'}$, and $A_2 \in \mathbb{R}^{d' \times r_l}$, $B_2 \in \mathbb{R}^{r_l \times d''}$ are layer-specific low-rank matrices with a rank of $r_l$. Typically, we constrain $r = r_s + r_l$. The advantages of CoLoRA are at least three-fold: first, it emulates the correlated updates of gradient descent for sequential layers as described in Eqs. (3) and (4). Second, by explicitly incorporating pre-trained weights into the tuning update in Eq. (6), it effectively guides the optimization direction compared to vanilla LoRA. Third, CoLoRA is potentially more parameter-efficient than vanilla LoRA. Specifically, LoRA requires a total of $r(d + 2d' + d'')$ parameters, whereas CoLoRA introduces $r_s(d + d'') + r_l(d + 2d' + d'')$ parameters, yielding a saving of $2r_s d'$ trainable parameters. For instance, using a ConvNeXt backbone where $d' = 4d = 4d''$, CoLoRA achieves a reduction of $8r_s d$ parameters (40% when $r_s = d/2$).

**Scaling factor**. For the independent updates outlined in Eq. (6), we find that the $s_{\text{expect}}$ discussed in section 3.1 is effective for both layers. Consequently, we utilize non-trainable scaling factors defined as $s_1 = 1/\sqrt{r_l}(\sigma_{A_1} + \sigma_{B_1})$ and $s_2 = 1/\sqrt{r_l}(\sigma_{A_2} + \sigma_{B_2})$. However, when it comes to the layer-sharing update, we notice that $s_{\text{expect}}$[2] can lead to divergent loss values when fine-tuning on datasets with significant domain gaps, such as Retinopathy sub-task in the VTAB-1k [50] benchmark. Therefore, we employ a variance-matching scaling factor $s_{\text{var}}$, as defined in section 3.1, for the shared low-rank matrices. Our experiments in image classification, object detection, and semantic segmentation indicate that the configuration of these scaling factors facilitates a stable tuning process while ensuring satisfactory performance.

---

[2] $B_s W_2^T$ and $W_1^T A_s$ in Eq. (6) denote alternative low-rank $B$ and $A$ matrices, respectively.

## 3.4 Integrating CoLoRA into ConvNets

Our CoLoRA framework is designed to work effectively with every pair of adjacent layers. In this section, we will detail the integration of CoLoRA into standard ConvNets. Specifically, we aim to tackle two critical challenges: (1) the spatial issue of incorporating LoRA into convolutional kernels, which refers to the detrimental gradients that can backpropagate from local irrelevant regions to the LoRA weights, as illustrated in Figure 4, and (2) determining the proper locations for applying CoLoRA within a ConvNet, such as a vanilla ResNet.

**The spatial issue of LoRA in ConvNets.** The convolution kernel, denoted as $W \in \mathbb{R}^{d \times d' \times k \times k}(k > 1)$, consists of $d$ input channels and $d'$ output channels and operates on local $k \times k$ regions. During the forward pass of a ConvNet, the convolution kernel is applied across each local region in a sliding-window fashion. Consequently, the gradients are backpropagated from all local $k \times k$ regions of the output feature to the LoRA weights, as highlighted by the green and red boxes in Figure 4. However, many of these local regions may contain irrelevant information as underlined in the red boxes in Figure 4. Gradients backwarded from such regions can interfere with the adaptation process.

Enlarging the receptive fields of the LoRA weights is a straightforward approach to reduce the impact of irrelevant gradients [32]. However, implementing this naively can lead to an excessive number of tunable parameters, scaling as $\mathcal{O}(rdk^2)$. In this section, we propose a refined edge-preserving filtering technique that does not require any training. Given a feature map $F \in \mathbb{R}^{C \times H \times W}$, we employ channel-wise filtering as follows:

$$\tilde{F}_{c,h,w} = \frac{1}{Z_{c,h,w}} \sum_{(\delta h, \delta w) \in \Omega} k_s(\delta h, \delta w) k_r(F_{c,h,w}, F_{c,h+\delta h,w+\delta w}) F_{c,h+\delta h,w+\delta w}, \tag{7}$$

where $k_s(\cdot, \cdot)$ represents a spatial filtering kernel, such as a Gaussian kernel $e^{-(\delta h^2 + \delta w^2)/(2\sigma_s^2)}$ with variance $\sigma_s^2$, which can be computed in parallel across all spatial windows $\Omega$. On the other hand, $k_r(\cdot, \cdot)$ denotes a range kernel, which typically cannot be processed in parallel. Drawing inspiration from the work of [51], which uses a raised cosine function for $\mathcal{O}(1)$ Bilateral filtering, we adopt a simple range kernel defined as $k_r(x, y) = \cos(\gamma_c(x - y)) = \cos(\gamma_c x)\cos(\gamma_c y) + \sin(\gamma_c x)\sin(\gamma_c y)$. This formulation allows us to compute spatial filtering on $\cos(\gamma_c F) \odot F$ and $\sin(\gamma_c F) \odot F$ to achieve edge-preserving filtering that expands receptive fields while suppressing noise within features. Further details are available in Appendix E. Note that $\gamma_c$ is a channel-wise scaling factor designed to constrain $\gamma_c(F_{c,h,w} - F_{c,h+\delta h,w+\delta w}) \in$

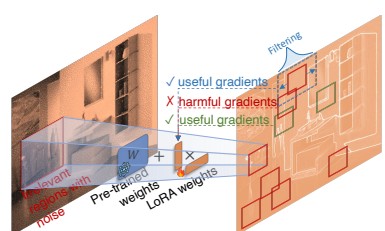

Figure 4: Spatial issues arising from applying LoRA to convolution kernels.

$[-\pi/2, \pi/2]$, ensuring that the range kernel applies greater weights to similar pixels, which helps preserve distinct edges. Specifically, we calculate $\gamma_c$ as $\frac{\pi}{2(\max F_c - \min F_c + \epsilon)}$, where $\epsilon = 10^{-5}$ is introduced to avoid division by zero. In practice, we apply our filtering method to features compressed by the low-rank matrix $A$ to curtail the computational burden associated with this filtering technique.

**Where to apply CoLoRA?** We adapt CoLoRA to mimic the weight update behavior of two consecutive layers. ConvNets such as ResNet-50 [52], consist of numerous sequential convolutional layers, and they are constructed from fundamental residual blocks. In this context, we implement our CoLoRA within each residual block by pairing every two adjacent convolutional layers from the bottom up, while using vanilla LoRA on the remaining layer that cannot be paired.

## 4 Experiments

**Setup.** We investigate the effectiveness of the proposed CoLoRA on two families of ConvNets: ResNet [52] and the more contemporary ConvNeXt [17]. For fine-tuning these ConvNets on various downstream tasks, we utilize the official ImageNet-22K pre-trained ResNet and ConvNeXt model series. Our evaluation encompasses a range of vision tasks, including classification on the large-scale visual adaptation benchmark VTAB-1k [50], object detection on the MS-COCO [53] dataset, and segmentation on the ADE20K [20] benchmark. Notably, VTAB-1k comprises 19 tasks that span a diverse array of categories, including seven NATURAL, four SPECIALIZED, and eight STRUC-

Table 1: Average top-1 accuracy on the VTAB-1k benchmark of three runs (See Appendix I for details on standard deviation). The best results among PEFT methods are in **bold**.

| Backbone | Method | #Param | Caltech101 | CIFAR-100 | DTD | Flowers102 | Pets | Sun397 | SVHN | Camelyon | EuroSAT | Resisc45 | Retinopathy | Clevr-Count | Clevr-Dist | DMLab | dSpr-Loc | dSpr-Ori | KITTI-Dist | sNORB-Azim | sNORB-Elev | Average |
|---|---|---|---|---|---|---|---|---|---|---|---|---|---|---|---|---|---|---|---|---|---|---|
| ResNet-50 | FT | 23.5M | 89.49 | 30.82 | 64.41 | 89.35 | 84.43 | 31.00 | 82.51 | 85.31 | 91.83 | 80.94 | 73.82 | 43.27 | 55.21 | 45.67 | 80.18 | 41.93 | 80.59 | 26.54 | 48.21 | 64.50 |
| | Conv-Adapter [32] | 1.4M | 86.98 | 27.22 | 64.40 | 82.21 | 88.98 | 32.67 | 51.31 | 78.59 | 88.20 | 75.29 | 73.80 | 35.94 | 44.98 | 35.40 | 41.50 | 15.29 | 69.95 | 14.72 | 38.21 | 55.03 |
| | PiSSA [7] | 1.2M | 88.12 | 26.74 | 62.93 | 85.85 | 89.13 | **32.71** | 63.32 | 81.59 | 90.25 | 78.47 | 73.82 | 48.61 | 49.16 | 38.21 | 52.62 | 22.29 | 73.70 | 18.05 | 39.43 | 58.68 |
| | HiRA [22] | 1.2M | 87.00 | 27.97 | **63.99** | 82.39 | **89.20** | 32.28 | 53.31 | 79.07 | 89.10 | 75.42 | 73.85 | 40.98 | 46.69 | 35.10 | 45.70 | 16.94 | 71.31 | 13.53 | 44.05 | 56.20 |
| | CoLoRA (ours) | 1.0M | **89.12** | **29.98** | 62.82 | **87.51** | 88.87 | 32.38 | **75.31** | **82.60** | **92.16** | **81.08** | **74.30** | **54.04** | **53.95** | **42.05** | **76.42** | **37.18** | **79.98** | **22.58** | **48.81** | **63.74** |
| ConvNeXt-S | FT | 49.5M | 91.33 | 65.60 | 74.17 | 98.74 | 90.36 | 49.02 | 92.30 | 87.95 | 95.98 | 85.98 | 76.83 | 91.06 | 66.58 | 55.10 | 92.75 | 62.41 | 83.88 | 39.96 | 46.91 | 76.15 |
| | Conv-Adapter | 2.8M | 90.94 | 68.52 | 75.37 | 99.12 | 91.13 | 49.91 | 90.35 | 85.75 | 94.89 | 84.04 | 75.30 | 87.82 | 66.06 | 48.68 | 94.11 | 61.30 | **85.04** | 37.29 | 49.14 | 75.51 |
| | PiSSA [7] | 3.0M | 90.93 | 69.22 | 75.62 | 99.06 | 91.29 | 51.96 | 90.27 | 85.79 | 95.45 | 85.56 | 75.56 | 91.51 | **67.04** | 51.35 | **94.47** | 61.47 | 82.79 | 37.24 | 49.15 | 76.09 |
| | HiRA | 3.0M | 90.71 | **70.85** | **76.30** | 99.27 | **92.12** | **53.56** | 86.39 | 84.29 | 94.85 | 84.91 | 75.85 | 79.64 | 63.95 | 47.33 | 79.60 | 56.35 | 82.32 | 25.85 | 42.13 | 72.96 |
| | CoLoRA (ours) | 2.6M | **91.60** | 66.34 | 75.00 | 98.89 | 90.43 | 49.94 | **92.08** | **87.51** | **96.00** | **85.95** | **77.13** | **91.71** | 65.57 | **54.59** | 92.94 | **62.28** | 83.82 | **40.71** | 48.95 | **76.39** |

TURED tasks. These benchmarks provide a comprehensive study for evaluating the proposed CoL-oRA in comparison to state-of-the-art techniques. All experiments are conducted on NVIDIA A800 GPUs using the PyTorch [54] framework, supported by APEX [55] for mixed precision training.

We compare CoLoRA with several recent PEFT methods: 1) Conv-Adapter [32], which is specifically designed for ConvNets and incorporates lightweight adapters into intermediate $K \times K$ convolution layers ($K > 1$). 2) PiSSA [7], which introduces an improved LoRA initialization scheme by decomposing pre-trained weights into principal tunable components and frozen residual parts. 3) HiRA [22], the latest high-rank tuning method, which constructs high-rank weight updates from low-rank matrices using the Hadamard product, i.e., $\Delta W = W \odot (BA)$. To ensure a fair comparison, we align the number of tunable parameters across different methods by adjusting the ranks of $B$ and $A$. Specifically, we define the rank $r$ as $r = C/\gamma$, following [32], where $C$ represents the channel number and $\gamma$ is a compression factor. Moreover, we adopt existing training-free LoRA merging techniques [24, 56] to enhance the adaptation ability of CoLoRA. Further details on complexity analysis and training configurations can be found in section 4.5, Appendix F, and Appendix H.

## 4.1 Evaluation on Vision Adaptation Benchmark

We begin by examining the effectiveness of CoLoRA on the VTAB-1k benchmark, utilizing ResNet-50 and ConvNeXt-S as backbones. We set the compression factor to $\gamma = 16$ and ensure that the ranks of the shared and layer-specific matrices are equal, i.e., $r_s = r_l$. This configuration results in approximately 1.0M trainable parameters for ResNet-50 and 2.6M for ConvNeXt-S, savings 0.2M and 0.4M parameters compared to PiSSA and HiRA, respectively. The average top-1 accuracy results from three runs are reported in Table 1, where CoLoRA demonstrates the highest adaptation performance among all PEFT methods. When using ResNet-50 as the backbone, CoLoRA achieves average accuracy gains of 8.71%, 5.06%, and 7.54% over Conv-Adapter, PiSSA, and HiRA, respectively. CoLoRA even surpasses full-tuning performance with ConvNeXt-S, while recent methods experience significant performance degradation. These results highlight the considerable advantages of mimicking the correlation of weight updates.

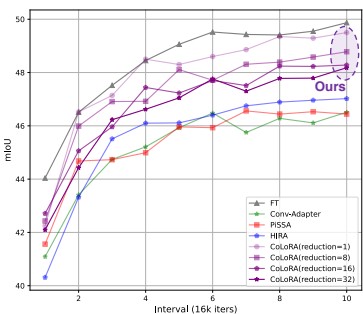

Figure 5: Evolution of segmentation mIoU with the 160k training schedule.

## 4.2 Comparison on Semantic Segmentation

We opt for ConvNeXt-S and ConvNeXt-B as backbones for our evaluations on semantic segmentation. Following the mainstream 160k training schedule on ADE20K [20], we utilize the MM-Segmentation [57] framework, employing UPerNet [58] for semantic prediction at a resolution of $512 \times 512$. The mean Intersection over Union (mIoU) metric is calculated using the default single-scale regime. While a multi-scale evaluation paradigm can enhance mIoU, it also incurs a greater computational burden. We denote the configuration with $\gamma = 16$ and $r_s = r_l$ as CoLoRA, and a stronger configuration with $\gamma = 8$ and $r_s = 4r_l$ to reduce parameters as CoLoRA$^{\dagger}$. Further

Table 2: Quantitative comparison of different PEFT methods on object detection (COCO) and semantic segmentation (ADE20K) tasks. The best results are in **bold**.

| Backbone | Method | #Param | Object Detection (COCO) | | | | Semantic Segmentation (ADE20K) |
|---|---|---|---|---|---|---|---|
| | | | AP$_{box}$@0.5 | AP$_{box}$@0.75 | AP$_{box}$ | AP$_{mask}$ | mIoU |
| | FT | 49.5M | 70.0 | 52.5 | 47.5 | 43.0 | 49.87 |
| | Conv-Adapter [32] (CVPRW'24) | 2.8M | 65.7 | 45.5 | 41.9 | 39.3 | 46.51 |
| ConvNeXt-S | PiSSA [7] (NeurIPS'24) | 3.0M | 65.8 | 45.8 | 41.8 | 39.6 | 46.56 |
| | HiRA [22] (ICLR'25) | 3.0M | 65.3 | 44.7 | 41.2 | 39.1 | 47.02 |
| | CoLoRA (ours) | **2.6M** | **66.8** | **47.9** | **43.6** | **40.3** | **48.28** |
| | FT | 87.6M | 71.4 | 54.2 | 49.1 | 44.1 | 50.99 |
| | Conv-Adapter [32] (CVPRW'24) | 6.3M | 68.7 | 49.2 | 44.9 | 41.2 | 48.73 |
| ConvNeXt-B | PiSSA [7] (NeurIPS'24) | 5.3M | 68.3 | 48.6 | 44.2 | 41.3 | 47.91 |
| | HiRA [22] (ICLR'25) | 5.3M | 67.6 | 47.6 | 43.5 | 40.8 | 48.41 |
| | CoLoRA (ours) | **4.6M** | 69.9 | 51.8 | 47.0 | 42.9 | 49.55 |
| | CoLoRA$^\dagger$ (ours) | 5.7M | **70.7** | **52.3** | **47.5** | **43.3** | **50.65** |

Table 3: Ablation studies on our contributions and designs. "Spatial Conv" denotes convolution layers with kernel size larger than 1, while "Channel Conv" specifies the $1 \times 1$ convolution layers.

| Backbone | Spatial Conv | Channel Conv | Correlated LoRA | Merging LoRA | mIoU |
|---|---|---|---|---|---|
| | LoRA | - | ✗ | ✗ | 46.30 |
| | LoRA + Filtering | - | ✗ | ✗ | 47.04 |
| ConvNeXt-S | LoRA + Filtering | LoRA | ✗ | ✗ | 47.14 |
| | LoRA + Filtering | LoRA | ✓ | ✗ | 48.00 |
| | LoRA + Filtering | LoRA | ✓ | ✓ | **48.28** |

details can be found in the Appendix H. A quantitative comparison is presented in Table 2, where all recent PEFT methods, including Conv-Adapter, PiSSA, and HiRA, experience significant performance degradation compared to the full-tuning scenario. Notably, our CoLoRA demonstrates an improvement of 1.26 and 0.82 mIoU over the previous best methods with ConvNeXt-S and ConvNeXt-B backbones, respectively. Additionally, our CoLoRA$^\dagger$, with slight 1.1M extra parameters, achieves an mIoU of 50.65, surpassing the current top-performing method by 1.92 mIoU, almost comparable to full-tuning. Remarkably, we achieve 51.13 mIoU with only 19M tunable parameters ($\gamma = 4, r_s = 2/3r_l$), thereby slightly outperforming full-tuning by an impressive 0.14 mIoU gap. More results are available in the Appendix J. These findings underscore the superiority of our CoLoRA and its considerable potential to match or even exceed full-tuning performance. Figure 5 illustrates the evolution of mIoU throughout the entire tuning process of various methods using ConvNeXt-S. Our CoLoRA consistently achieves higher mIoU during the entire tuning process, indicating superior convergence compared to other methods.

## 4.3 Experiments on Object Detection

In this section, we present experiments on object detection utilizing the standard Faster R-CNN framework [59]. We select ConvNeXt-S and ConvNeXt-B as our backbone architectures. All methods are implemented within the MMDetection framework [60], adhering to a standard 1x schedule over 12 epochs. As in section 4.2, we train two variants of the proposed method: CoLoRA and CoLoRA$^\dagger$. The performance metrics include $AP_{email_1}$, $AP_{email_2}$, $AP_{box}$, and $AP_{mask}$, as detailed in Table 2. Typically, we adopt a single-scale evaluation regime for metric computation. From Table 2, it is evident that Conv-Adapter, PiSSA, and HiRA experience significant performance degradation compared to full tuning. For instance, using ConvNeXt-B as the backbone, the high-rank tuning method, HiRA, shows a decline of 5.6 in $AP_{box}$. Even with a more effective LoRA initialization, PiSSA fails to deliver satisfactory performance. Conversely, incorporating adapters into convolution layers, specifically Conv-

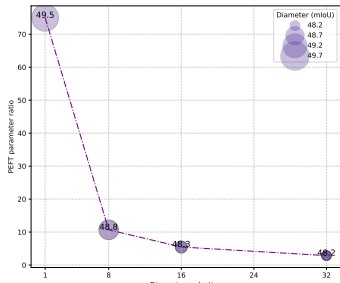

Figure 6: Investigation of the dimension compression factor.

Adapter, yields only marginal improvements over PiSSA and HiRA. In contrast, our CoLoRA results in a notable enhancement in performance. Particularly, CoLoRA$^\dagger$ achieves a 2.6 $AP_{box}$ gain over the best-performing method, Conv-Adapter, while also reducing the parameter count by 0.6M. These findings underscore the significance of emulating correlated weight updates in fine-tuning ConvNets.

## 4.4 Extend CoLoRA to Vision Transformer on Fine-grained Classification

We further extend the proposed CoLoRA framework to the Vision Transformer (ViT) architecture by fine-tuning a pre-trained ViT-B model initialized with MAE [3]. The results, summarized in Table 4, present top-1 accuracies on two fine-grained classification benchmarks, CUB-200-2011 [61] and FGVC-Aircraft [62]. As shown, CoLoRA consistently outperforms PiSSA and HiRA, demonstrating superior adaptability within transformer-based architectures.

## 4.5 Complexity Analysis

To mitigate the computational overhead introduced by the filtering strategy in Eq. (7), we apply `torch.jit.script` for operator-level optimization and omit gradient computations from the trigonometric kernels in Eq. (7) to simplify the backward pass. A comprehensive complexity analysis comparing PiSSA, HiRA, and CoLoRA is presented in Table 5. All LoRA-based variants demonstrate reduced training memory consumption and shorter training time

Table 4: Quantitative results of tuning a ViT-B backbone on two fine-grained classification benchmarks, CUB-200-2011 and FGVC-Aircraft.

| Datasets | FT | PiSSA | HiRA | CoLoRA |
|----------|-----|-------|------|--------|
| CUB-200-2011 | 79.24 | 74.70 | 51.97 | 75.61 |
| FGVC-Aircraft | 79.30 | 72.40 | 34.32 | 72.79 |

compared to full fine-tuning (FT). Notably, the additional cost introduced by the bilateral filtering module is negligible (e.g., 1.509s vs. 1.527s per iteration). It is also observed that, under mixed-precision training, removing the filtering module may slightly increase GPU memory usage due to precision conversion overhead. During inference, the weight-merging property of LoRA ensures identical runtime across all variants, indicating that the filtering mechanism in CoLoRA does not compromise inference efficiency. Collectively, these observations validate the computational efficiency and practical feasibility of the optimized CoLoRA framework.

Table 5: Training and inference complexity analysis of the proposed CoLoRA, evaluated on the MS-COCO dataset using a ConvNeXt-S backbone.

| Methods | FT | PiSSA | HiRA | CoLoRA | CoLoRA (w/o filtering) |
|---------|-----|-------|------|--------|------------------------|
| Training GPU memory (MiB) | 18418 | 14648 | 14536 | 15042 | 16488 |
| Per-step training time (s) | 1.747 | 1.426 | 1.501 | 1.527 | 1.509 |
| Inference time (s) | 0.108 | 0.108 | 0.108 | 0.108 | 0.108 |

## 4.6 Ablation Studies

In this section, we perform ablation studies to validate the effectiveness of our contributions and designs. All experiments are conducted using the ADE20K semantic segmentation benchmark, featuring ConvNeXt-S as our backbone. We use ConvNeXt-S with LoRA applied exclusively to the depth convolution layers as our baseline, as suggested by a recent study [32].

**Does filtering improve performance?** We incorporate our filtering method detailed in Eq. (7) into the baseline. As indicated in Table 3, this adjustment increases the mIoU from 46.30 to 47.04, yielding a gain of 0.74 mIoU. This finding demonstrates that expanding the receptive field of LoRA weights positively influences the tuning of ConvNets.

**Where should low-rank adaptation be applied?** Recently, Conv-Adapter [32] proposed tuning solely the spatial convolution layers while leaving the $1 \times 1$ convolution layers untouched. Notably, adjacent $1 \times 1$ convolution layers operate as an inverted bottleneck design within the ConvNeXt [17] architecture. This design through MobileNetV2 [18] has been widely explored in advanced ConvNets [19, 63]. We hypothesize that tuning the $1 \times 1$ convolution layers could yield additional performance gains. However, as illustrated in Table 3, applying LoRA to these layers results in only a modest 0.1 mIoU improvement, which aligns with the results of Conv-Adapter. To explore this contradiction further, we conducted a full tuning of all $1 \times 1$ convolution layers in ConvNeXt-S while keeping the other layers frozen. This configuration achieves an mIoU of 50.45, exceeding the performance of full-tuning (49.87 in Table 2). Hence, we conclude that *the $1 \times 1$ convolution lay-*

*ers exhibit significant potential for tuning. However, this potential cannot be effectively harnessed through layer-independent LoRA or adapters.*

**Impact of correlated weight updates**. We show that the potential of the inverted bottleneck can be effectively harnessed through the proposed correlated weight update method (as detailed in section 3.3). As shown in Table 3, we improve the mean mIoU from 47.14 to 48.00 using the correlated LoRA update. We achieve an impressive mIoU of 50.65 while only tuning 5.7M parameters, outperforming the Conv-Adapter, which still lags behind full-tuning despite having approximately 50% more tunable parameters (39.40M; refer to [32] for further details).

**Impact of Periodically Merging LoRA Weights**. Figure 1 illustrates that tuning pre-trained models for downstream tasks closely resembles a nearly full-rank learning process. In this study, we leverage the current LoRA merging technique [24] to enhance the rank of $\Delta W$. Specifically, we periodically merge the weight updates computed via Eq. (6) into the pre-trained weights,and subsequently re-initialize the LoRA matrices. Following the suggestions of [24], we reset the optimizer and implement a restart warmup schedule to stabilize the tuning process. This approach yields a noticeable improvement of 0.28 mIoU on ConvNeXt-S, as depicted in Table 3.

**Analysis on various dimension compression factors**. We analyze the effect of the compression factor, $\gamma$. A smaller value of $\gamma$ generally results in higher-rank weight updates with an increased number of tunable parameters. Figure 6 visualizes the ratio of tunable parameters alongside the mIoU metric for different $\gamma$. Notably, full-rank tuning can be achieved with a tunable parameter ratio of 75% ($< 1$) due to the phenomenon of parameter sharing. Empirically, this full-rank tuning yields a segmentation mIoU of 49.5, which is comparable to the full-tuning. Furthermore, even at a compression factor of $\gamma = 32$, the proposed CoLoRA framework surpasses the performance of Conv-Adapter by a margin of 1.67 mIoU, despite utilizing only 1.4M tunable parameters, which is less than half of the parameters for Conv-Adapter. When progressively reducing $\gamma$ from 32 to 8, we consistently observe an increase in mIoU from 48.2 to 48.8. Moreover, maintaining $\gamma$ within the range of [1, 32] systematically results in enhanced convergence when compared to state-of-the-art methods as shown in Figure 5.

**Analysis on the rank of $A_s$ and $B_s$**. We explore how performance metrics vary under different correlation strengths as delineated in Eq. (6). For our experiments, we maintain a fixed compression factor at $\gamma = 16$ while modifying the ratio $\frac{r_s}{r_s+r_l}$. An increase in this ratio signifies a stronger correlation. As demonstrated in Table 6, we identify that the exclusive correlation of LoRA updates produces a mean mIoU of 47.60. The optimal mIoU is attainable through the combined application of both correlated and layer-specific LoRA updates, as this strategy effectively emulates the gradient behavior outlined in Eq. (4).

Table 6: Effect of $r_s$.

| $\frac{r_s}{r_s+r_l}$ | 0 | 50% | 1 |
|---|---|---|---|
| mIoU | 47.14 | **48.28** | 47.60 |

## 5   Conclusion

This paper presents CoLoRA, an innovative and highly PEFT methodology designed for the adaptation of pre-trained ConvNets across a variety of downstream applications. Rather than solely incorporating independent LoRA to each layer in ConvNets, the proposed method focuses on emulating the correlated weight update behavior between adjacent layers. In addition, we introduce a parameter-free filtering strategy aimed at the seamless integration of correlated weight updates within ConvNets. Through extensive experiments encompassing large-scale classification, semantic segmentation, and object detection, we demonstrate that CoLoRA significantly outperforms existing PEFT approaches.

## 6   Limitations

Despite its effectiveness, CoLoRA still has several limitations. First, this work primarily focuses on adapting pre-trained vision backbones for various downstream vision tasks, leaving its potential on emerging vision-language model-based perception tasks, such as visual question answering and visual grounding unexplored. In addition, this study mainly investigates 2D vision scenarios. Extending CoLoRA to 3D domains may introduce new challenges and opportunities, further broadening the scope and impact of this research.

**Acknowledgements**. This work was supported in part by NSFC (62322113, 62376156), Shanghai Municipal Science and Technology Major Project (2021SHZDZX0102), and the Fundamental Research Funds for the Central Universities.

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

# Appendix

## A    Proof of Lemma 3.1

Suppose all elements in $A \in \mathbb{R}^{d \times r}$ are i.i.d., initialized with the uniform distribution $\mathcal{U}(-a, a)$. Typically, we have $\mathbb{E}[A_{ij}] = 0$ and $\text{Var}[A_{ij}] = \mathbb{E}[A_{ij}^2] = \frac{1}{3}a^2$. Notably, we have $\sigma_{A_{ij}}^2 = \frac{1}{3}a^2$. Denote the matrix multiplication result as $T = AA^T$; then we can calculate

$$\mathbb{E}[T_{ij}] = \mathbb{E}[\sum_{k=1}^{r} A_{ik}A_{jk}] = \sum_{k=1}^{r} \mathbb{E}[A_{ik}A_{jk}] = \begin{cases} 0 & i \neq j, \\ r\sigma_A^2. & \text{else} \end{cases} \tag{8}$$

Therefore, we have $\mathbb{E}[AA^T] = r\sigma_A^2 \mathbf{I}_d$, where $\mathbf{I}_d$ denotes the $d \times d$ identity matrix. Next, we analyze the variance of the elements in $T$. For the off-diagonal elements of $T$, we have

$$\text{Var}[T_{ij}] = \sum_{k=1}^{r} \mathbb{E}[A_{ik}^2 A_{jk}^2] = r\sigma_A^4, s.t., i \neq j. \tag{9}$$

For the diagonal elements, we have

$$\text{Var}[T_{ii}] = \sum_{k=1}^{r} \mathbb{E}[A_{ik}^4] - \mathbb{E}[A_{ik}^2]^2 = \frac{4}{5}r\sigma_A^4, \tag{10}$$

where we can compute $\mathbb{E}[A_{ik}^4] = \frac{1}{a}\int_0^a x^4 \mathbf{d}x = \frac{1}{5}a^4 = \frac{9}{5}\sigma_A^4$. Summarizing the above results, we obtain the variance of $AA^T$ as

$$\text{Var}[T_{ij}] = \begin{cases} r\sigma_A^4 & i \neq j, \\ \frac{4}{5}r\sigma_A^4 & \text{else.} \end{cases} \tag{11}$$

In this paper, we approximate $\text{Var}[T] \approx r\sigma_A^4 \mathbf{1}_d$, where $\mathbf{1}_d$ means a $d \times d$ matrix with all elements being 1. The approximation error in terms of the Frobenius norm can be formed as

$$\text{Err} = ||\text{Var}[T] - r\sigma_A^4 \mathbf{1}_d||_F^2 = \frac{1}{25}r^2 d\sigma_A^8. \tag{12}$$

Note that with `kaiming_uniform` initialization where $a \propto 1/\sqrt{d}$, the approximation error $\text{Err} \propto r^2/d^3$. Hence, we have

$$\frac{\text{Err}}{||\text{Var}[T]||_F^2} \approx \frac{1}{25} \frac{r^2/d^3}{d^2 r^2/d^4} = \frac{1}{25d}. \tag{13}$$

It is evident that $\text{Err}/||\text{Var}[T]||_f^2$ is independent of the rank $r$ and is inversely proportional to the channel dimension $d$. When $d$ is large (e.g., 96, 192, 384, 768 for ConvNeXt-B), we can approximate $\text{Var}[AA^T] \approx r\sigma_A^4$ with a maximum relative error of $1/(25 \times 96) \approx 0.04\%$.

## B    Tuning with Various LoRA Ranks

To provide a more rigorous justification for the high-rank tuning strategy of LoRA, we conduct experiments by fine-tuning ConvNeXt-B using PiSSA under varying ranks. For convolutional networks, the rank is parameterized by a channel compression factor $\gamma$, where $r = d/\gamma$ and $d$ denotes the number of channels.

Notably, $\gamma = 1$ corresponds to full-rank adaptation. It is worth mentioning that LoRA introduces over-parameterization when $r = 1$ due to the decomposition $\Delta W = sBA$, resulting in more trainable parameters than the original full-rank weight. Quantitative results are summarized in Table 7. Specifically, performance peaks at $r = 2$, while both under- and over-parameterization lead to degradation. Increas-

Table 7: Tuning ConvNeXt-B on ADE20K with different ranks of PiSSA.

| $\gamma$ | 1 | 2 | 4 | 8 | 20 |
|---|---|---|---|---|---|
| #Params (tunable) | 104.8M | 52.5M | 26.3M | 13.2M | 5.3M |
| Param ratio (tunable) | 120% | 60% | 30% | 15% | 6% |
| mIoU | 50.1 | 50.6 | 49.9 | 49.1 | 47.6 |

ing $\gamma$ consistently reduces mIoU from 50.6 to 47.6. These findings validate that independent LoRA tuning on ConvNeXt-B benefits from high-rank adaptation, whereas performance deteriorates under tight parameter constraints. This analysis will be incorporated into the revised manuscript to further substantiate our argument.

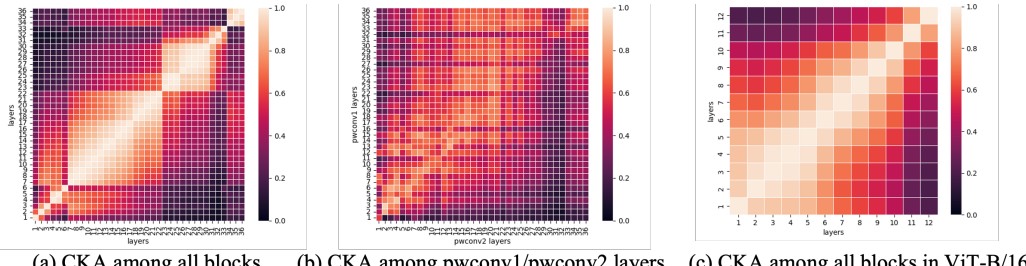

| (a) CKA among all blocks | (b) CKA among pwconv1/pwconv2 layers | (c) CKA among all blocks in ViT-B/16 |

Figure 7: CKA analysis for ConvNeXt-B and ViT-B/16. (a) shows CKA similarities among all 36 blocks. (b) presents CKA similarties among all 36 `pwconv1` and `pwconv2` layers. (c) examines CKA similarities among all 12 blocks within ViT-B/16.

## C    More Evidence on Weight Update Correlation in ConvNets

In this section, we provide additional evidence to investigate the correlation between adjacent convolution layers. Specifically, we select the ConvNeXt backbone and examine the correlation between the adjacent `pwconv1` and `pwconv2` layers. We denote their convolution weights as $W_1$ and $W_2$, respectively. To demonstrate that the updates of $W_1$ and $W_2$ are highly correlated, we compute $\Delta W$ by subtracting the old weight $W_{\text{old}}$ from the new weight $W_{\text{new}}$, i.e., $\Delta W = W_{\text{new}} - W_{\text{old}}$. Typically, $W_{\text{old}}$ and $W_{\text{new}}$ correspond to the weights saved from the previous and current epochs, respectively. For ADE20K, model weights are checkpointed every 16K iterations, while for MS-COCO, weights are saved after every epoch.

Since both $W_1$ and $W_2$ exhibit full-rank update behavior, we assess their correlation by computing the mean of the singular values of $\Delta W_1$ and $\Delta W_2$. If their mean singular values exhibit similar trends throughout the fine-tuning process, it indicates that the update weight matrices for $W_1$ and $W_2$ are highly correlated.

Figure 8 presents the mean singular values of $\Delta W_1$ and $\Delta W_2$ throughout the fine-tuning process on ADE20K. Specifically, we analyze the weights of the `pwconv1` and `pwconv2` layers across all residual blocks in ConvNeXt-B. It can be observed that nearly all adjacent `pwconv1` and `pwconv2` layers exhibit strong correlation in their update behavior.

Figure 9 shows the mean singular values of $\Delta W_1$ and $\Delta W_2$ throughout the fine-tuning process on MS-COCO. Specifically, we analyze the weights of the `pwconv1` and `pwconv2` layers across all residual blocks in ConvNeXt-S. It can be observed that nearly all adjacent `pwconv1` and `pwconv2` layers exhibit strong correlation. The sharp drop in mean singular values is attributed to the learning rate scaling in the standard $1\times$ schedule. The results in Figures 8 and 9 collectively demonstrate that adjacent convolution layers exhibit high correlation during fine-tuning on downstream tasks such as semantic segmentation and object detection.

## D    Centered Kernel Alignment Analysis

In this section, we further investigate the correlation between adjacent layers or blocks within ConvNets and ViTs. We employ the Centered Kernel Alignment (CKA) metric [64] to quantify the similarity between specific layer or block pairs. The CKA metric offers the following advantages:

- CKA is invariant to orthogonal transformations and scaling.
- CKA remains applicable even when the compared features have different dimensionalities, whereas metrics such as the Frobenius inner product cannot be used in such cases.
- CKA has been widely adopted to analyze the representational behaviors of ConvNets and Vision Transformers (ViTs) [21].

We first select a pre-trained ConvNeXt-B model consisting of 36 convolutional blocks. As shown in Figure 7(a), the CKA scores across all blocks reveal that long-range block pairs exhibit lower similarity, whereas adjacent blocks consistently show higher similarity. This indicates that nearby

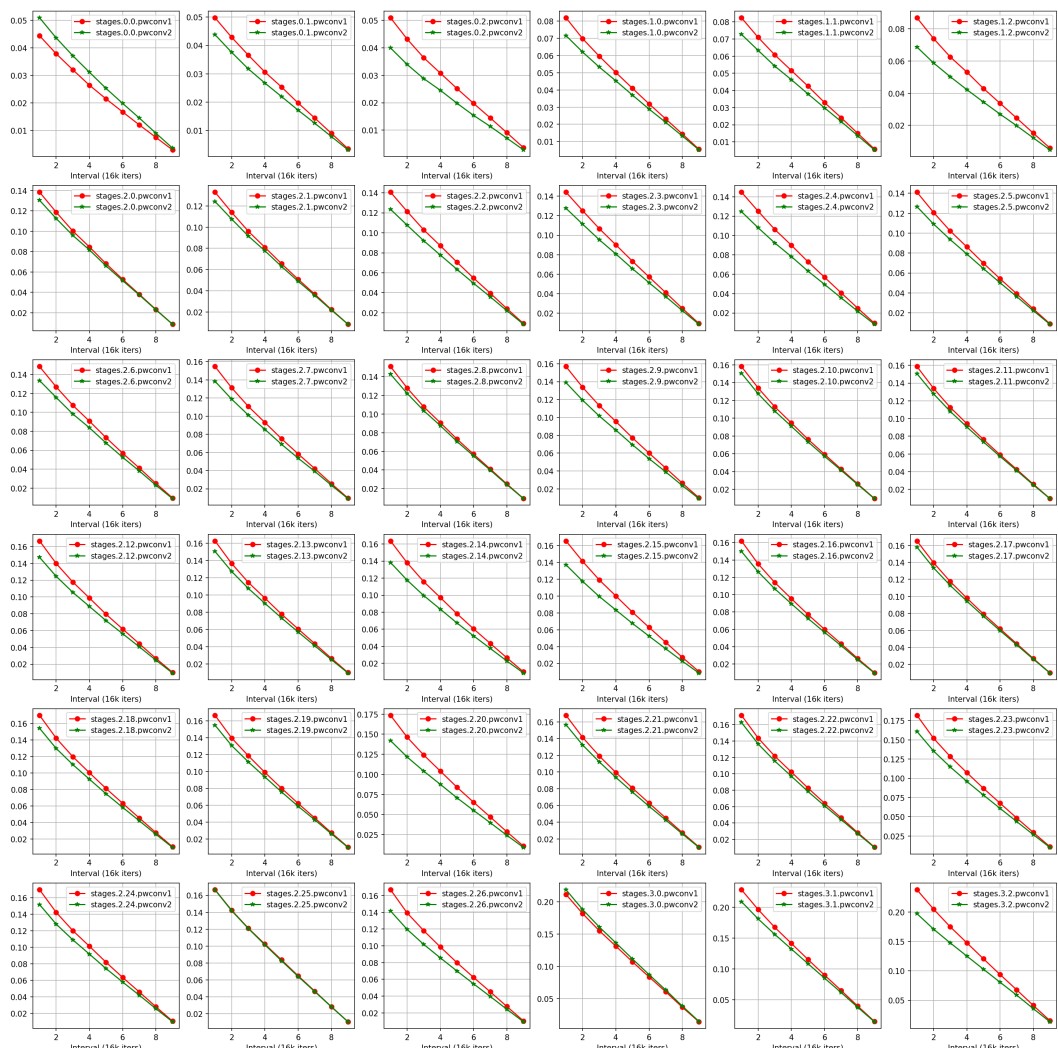

Figure 8: Correlation between `pwconv1` and `pwconv2` layers in all 36 residual blocks. We fully fine-tune the pre-trained ConvNeXt-B on ADE20K.

convolutional blocks are strongly correlated. Furthermore, we employ CKA to analyze the similarity between two adjacent convolutional layers `pwconv1` and `pwconv2` as illustrated in Figure 7(b). The results show that these paired layers exhibit high similarity, suggesting strong correlation between adjacent convolutional layers. Based on these findings, we infer that such inter-layer correlations persist during downstream fine-tuning as well.

We extend the CKA to a pre-trained ViT-B/16 model and provide the result in Figure 7(c). It is obvious that the correlation strength within Transformer blocks are weaker than convolution blocks, suggesting lower correlation strength at the fine-tuning process.

## E   Details of Filtering-based Strategy

We first recall in Section 3.4 that the filtering process can be formulated as

$$\tilde{F}_{c,h,w} = \frac{1}{Z_{c,h,w}} \sum_{(\delta h, \delta w) \in \Omega} k_s(\delta h, \delta w) k_r(F_{c,h,w}, F_{c,h+\delta h,w+\delta w}) F_{c,h+\delta h,w+\delta w}, \quad (14)$$

where $k_s(\delta h, \delta w)$ denotes a spatial kernel, whereas $k_r(F_{c,h,w}, F_{c,h+\delta h,w+\delta w})$ represents a range kernel in the context of image filtering. Without the range kernel $k_r$, the above Eq. (14) can be

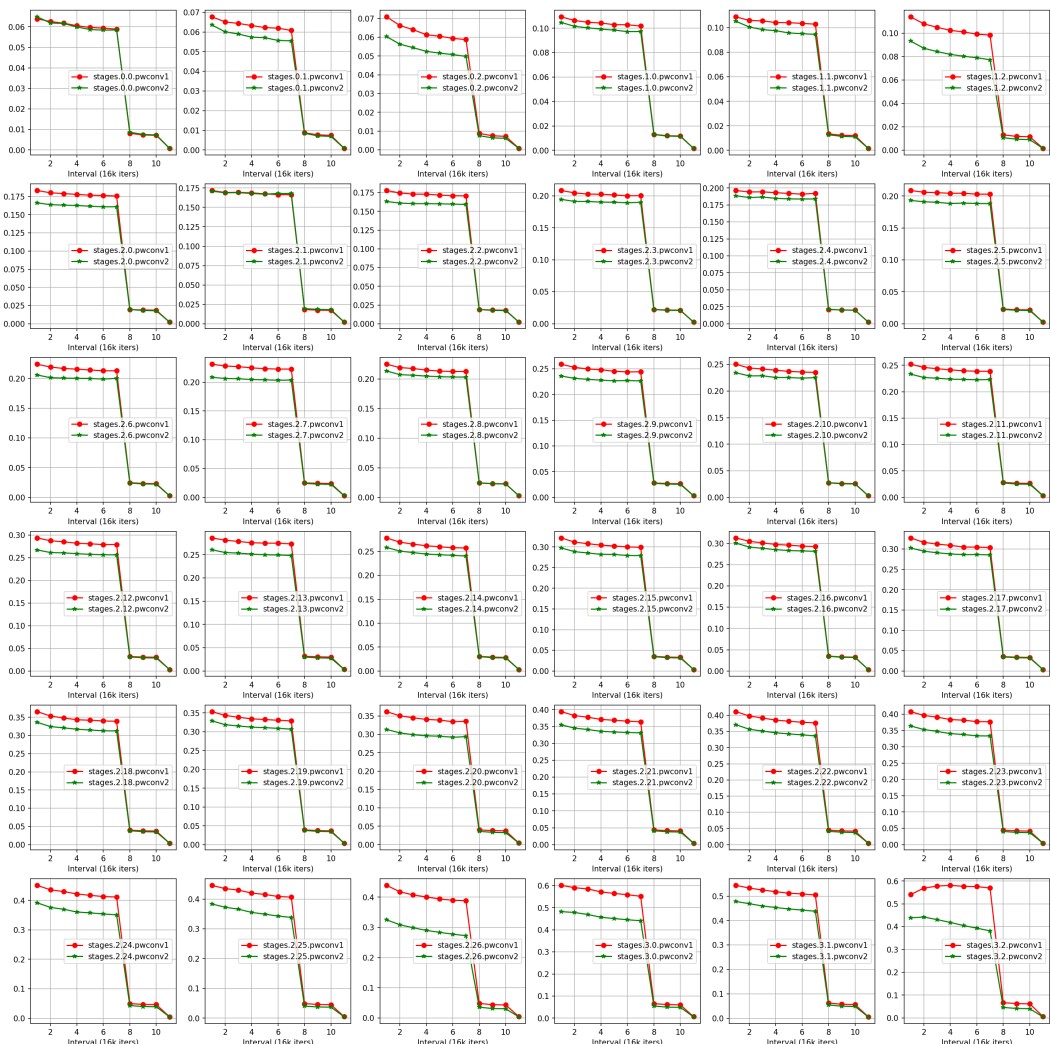

Figure 9: Correlation between `pwconv1` and `pwconv2` layers in all 36 residual blocks. We fully fine-tune the pre-trained ConvNeXt-S on MS-COCO.

simplified as

$$\tilde{F}_{c,h,w} = \frac{1}{Z_{c,h,w}} \sum_{(\delta h, \delta w) \in \Omega} k_s(\delta h, \delta w) F_{c, h+\delta h, w+\delta w} = \left[ \text{Conv}\left( \frac{k_s}{Z_c}, F_c \right) \right]_{h,w}, \qquad (15)$$

where the normalization factor $Z_c = \sum_{\delta h, \delta w} k_s(\delta h, \delta w)$ is spatially invariant. $\text{Conv}(K, X)$ means a convolution operation on $X$ utilizing a kernel $K$. Eq. (15) indicates that a channel-wise filtering on deep features can be expressed as a depthwise convolution operation, which can be rapidly computed with the help of existing algorithm.

However, with the range kernel $k_r(\cdot, \cdot)$, Eq. (14) requires a double-loop process through all $(h, w)$ coordinates. In this paper, we utilize a special range kernel $k_r(x, y) = \cos(\gamma_c(x - y)) = \cos(\gamma_c x)\cos(\gamma_c y) + \sin(\gamma_c x)\sin(\gamma_c y)$, where $\gamma_c$ is a constant to constrain that $\gamma_c(x - y) \in$

$[-\pi/2, \pi/2]$. Incorporating this range kernel into Eq. (14), we derive

$$
\tilde{F}_{c,h,w} = \frac{1}{Z_{c,h,w}} \sum_{(\delta h, \delta w) \in \Omega} k_s(\delta h, \delta w) k_r(F_{c,h,w}, F_{c,h+\delta h, w+\delta w}) F_{c,h+\delta h, w+\delta w},
$$

$$
= \frac{1}{Z_{c,h,w}} \sum_{(\delta h, \delta w) \in \Omega} k_s(\delta h, \delta w) \cos \gamma_c F_{c,h,w} \cos \gamma_c F_{c,h+\delta h, w+\delta w} F_{c,h+\delta h, w+\delta w}
$$

$$
+ \frac{1}{Z_{c,h,w}} \sum_{(\delta h, \delta w) \in \Omega} k_s(\delta h, \delta w) \sin \gamma_c F_{c,h,w} \sin \gamma_c F_{c,h+\delta h, w+\delta w} F_{c,h+\delta h, w+\delta w}
$$

$$
= \frac{[\cos \gamma_c F \odot \mathrm{Conv}(k_s, \cos \gamma_c F \odot F)]_{h,w} + [\sin \gamma_c F \odot \mathrm{Conv}(k_s, \sin \gamma_c F \odot F)]_{h,w}}{Z_{c,h,w}}, \quad (16)
$$

where the $\odot$ means the element-wise multiplication. Specifically, the normalization factor is

$$
Z_{c,h,w} = \sum_{(\delta h, \delta w) \in \Omega} k_s(\delta h, \delta w) \cos \gamma_c F_{c,h,w} \cos \gamma_c F_{c,h+\delta h, w+\delta w}
$$

$$
+ \sum_{(\delta h, \delta w) \in \Omega} k_s(\delta h, \delta w) \sin \gamma_c F_{c,h,w} \sin \gamma_c F_{c,h+\delta h, w+\delta w}
$$

$$
= [\cos \gamma_c F \odot \mathrm{Conv}(k_s, \cos \gamma_c F) + \sin \gamma_c F \odot \mathrm{Conv}(k_s, \sin \gamma_c F)]_{h,w}. \quad (17)
$$

Combining Eqs. (16) and (17), we obtain the final simplified form of Eq. (14) as follows:

$$
\tilde{F}_c = \frac{\cos \gamma_c F \odot \mathrm{Conv}(k_s, \cos \gamma_c F \odot F) + \sin \gamma_c F \odot \mathrm{Conv}(k_s, \sin \gamma_c F \odot F)}{\cos \gamma_c F \odot \mathrm{Conv}(k_s, \cos \gamma_c F) + \sin \gamma_c F \odot \mathrm{Conv}(k_s, \sin \gamma_c F) + \epsilon}, \quad (18)
$$

where we set $\epsilon = 10^{-5}$ to avoid division by zero. As shown in Eq. (18), only four depthwise convolution operations are required to compute Eq. (14), eliminating the need for time-consuming for-loops. For implementation, we employ a fixed Gaussian spatial kernel $k_s$ with a window size of 11 and a standard deviation of $\sigma = 1.0$. Consequently, Eq. (18) contains no trainable parameters, and gradients need not be computed for this filtering process.

Table 8: Complexity analysis of different LoRA methods. Here, $r$ denotes the rank of matrices $A$ and $B$, #Params represents the number of trainable parameters in LoRA matrices, and FLOPs measure the computational complexity of LoRA modulation during both training and inference phases.

| Method | Rank $r$ | #Params | FLOPs (train) | FLOPs (inference) |
|---|---|---|---|---|
| LoRA | $r$ | $r(d + 2d' + d'')$ | $(hwd' + rd' + d')(d + d'')$ | $hwd'(d + d'')$ |
| PiSSA | $r$ | $r(d + 2d' + d'')$ | $(hwd' + rd' + d')(d + d'')$ | $hwd'(d + d'')$ |
| HiRA | $r$ | $r(d + 2d' + d'')$ | $(hwd' + rd' + 2d')(d + d'')$ | $hwd'(d + d'')$ |
| CoLoRA (ours) | $r_l + r_s = r$ | $r(d + d'') + 2r_l d'$ | $(hwd' + rd' + 2d' + r_s d')(d + d'')$ | $hwd'(d + d'')$ |

## F Computational Complexity Analysis

The computational complexity of CoLoRA stems from both the proposed correlated low-rank adaptation process and the filtering technique.

**Complexity of correlated low-rank adaptation**. We analyze the complexity of two sequential $1 \times 1$ convolutional layers as described in Section 3. This analysis can be extended to arbitrary convolution layers. Consider an input feature $X \in \mathbb{R}^{hw \times d}$ (with spatial dimensions rearranged for simplicity), where $d$ denotes the channel dimension and $h, w$ represent its spatial resolution. The feature $X$ is processed by two weight matrices $W_1 \in \mathbb{R}^{d \times d'}$ and $W_2 \in \mathbb{R}^{d' \times d''}$. For clarity, we omit the analysis of bias terms as they are independent of LoRA. The vanilla LoRA modifies the feature $X$ through

$$
Y_1 = X(W_1 + s_1 A_1 B_1), \quad (19)
$$

$$
Y_2 = Y_1(W_2 + s_2 A_2 B_2), \quad (20)
$$

where $A_1 \in \mathbb{R}^{d \times r}$, $B_1 \in \mathbb{R}^{r \times d'}$, $A_2 \in \mathbb{R}^{d' \times r}$, and $B_2 \in \mathbb{R}^{r \times d''}$ are low-rank matrices with rank $r$, while $s_1$ and $s_2$ denote scaling factors. We contend that such layer-independent LoRA formulations fail to capture inter-layer correlations, which are particularly crucial for effectively fine-tuning convolutional networks. To overcome this limitation, we propose CoLoRA, which transforms the feature $X$ through

$$Y_1 = X(W_1 + s_1 A_1 B_1 + s A_s B_s W_2^T) \tag{21}$$

$$Y_2 = Y_1(W_2 + s_2 A_2 B_2 + s W_1^T A_s B_s), \tag{22}$$

where we incorporate shared low-rank matrices $A_s \in \mathbb{R}^{d \times r_s}$ and $B_s \in \mathbb{R}^{r_s \times d''}$ to explicitly model correlations between adjacent layers. To efficiently compute $A_s B_s W_2^T$ in Eq. (22), we first compute $C = B_s W_2^T$ followed by the product $A_s C$, leveraging the low-rank property $r_s \ll \min\{d, d', d''\}$. The computation of $W_1^T A_s B_s$ follows an analogous procedure. Table 8 compares the computational complexity of LoRA, PiSSA, HiRA, and our proposed CoLoRA.

During training, CoLoRA introduces an additional computational cost of $r_s d'(d + d'')$ FLOPs compared to existing HiRA methods. Note that since $r_s \ll \min\{hw, d, d', d''\}$, this overhead is negligible in practice. Furthermore, due to CoLoRA's weight decomposition formulation, we can directly incorporate the weight update matrices into the base weights $W_1$ and $W_2$ after training completion. Consequently, CoLoRA introduces no additional computational overhead during inference.

**Complexity of filtering technique**. As demonstrated in Eq. (18), the filtering operation can be efficiently implemented using four depthwise convolutions. To optimize computational efficiency, we apply the filtering process exclusively to features compressed by the LoRA matrix $A$. This design yields an overall computational complexity of $\mathcal{O}(hwk^2 r)$, where $k$ denotes the window size ($k = 11$ in our implementation).

## G   Details of Selected Pre-trained Backbones

In Table 9, we present the specifications of the pre-trained ConvNets backbones used in our study.

Table 9: Details of selected pre-trained backbones in this paper.

| Backbones | Pre-train dataset | #Params | Feature dim. | Public url |
|---|---|---|---|---|
| ResNet-50 | ImageNet-1K | 23.5M | 2048 | https://docs.pytorch.org/vision/main/models.html (v2) |
| ConvNeXt-S | ImageNet-22K | 49.5M | 768 | https://dl.fbaipublicfiles.com/convnext/convnext_small_22k_1k_384.pth |
| ConvNeXt-B | ImageNet-22K | 87.6M | 1024 | https://dl.fbaipublicfiles.com/convnext/convnext_base_22k_224.pth |

## H   Training Configurations

We conduct experiments by training ResNet-50, ConvNeXt-S, and ConvNeXt-B models on multiple downstream vision tasks. The detailed training configurations are provided in this section. The basic experimental configurations employed in our study are summarized in Tables 10 and 11. Notably, our implementation follows the official configurations from `https://github.com/facebookresearch/ConvNeXt`.

**Basic configuration**. The baseline configuration employs consistent hyperparameters across different downstream tasks and PEFT methods (with potential exceptions for MS-COCO, as detailed below). Tables 10 and 11 present the complete specifications of baseline hyperparameters and training configurations.

**Training configuration on VTAB-1k**. For VTAB-1k classification tasks, our processing pipeline consists of a Global Average Pooling (GAP) layer, followed by normalization and a linear classifier head. We optimize the models using standard cross-entropy loss. Each VTAB-1k sub-task is fine-tuned with a batch size of 32 for 100 epochs. In addition to the trainable low-rank matrices, we optimize all bias terms and normalization layers in the backbone network. Unlike existing approaches such as Conv-Adapter that perform grid search to identify optimal hyperparameters for individual sub-tasks, we maintain consistent hyperparameter settings across all sub-tasks. The specific configurations for each PEFT method are detailed below:

Table 10: Basic configuration for ResNet-50.

| Hyper-parameter | Setting |
| --- | --- |
| Optimizer | AdamW |
| Learning rate | 1e-4 |
| Weight decay | 0.05 |
| Layer weight decay | Number of layers is 16. Decay rate is 0.99 |
| Linear warmup | 1500 steps with ratio 1e-6 |
| Mixed precision training | APEX |

Table 11: Basic configuration for ConvNeXt (ConvNeXt-S and ConvNeXt-B).

| Hyper-parameter | Setting |
| --- | --- |
| Optimizer | AdamW |
| Learning rate | 1e-4 |
| Weight decay | 0.05 |
| Stage-wise decay | Number of stages is 4. Decay rate is 0.9 |
| Linear warmup | 1500 steps with ratio 1e-6 |
| Mixed precision training | APEX |

- Conv-Adapter. We utilize the official Conv-Adapter implementation from `https://github.com/Hhhhhhao/Conv-Adapter`. As Conv-Adapter exclusively applies trainable adapters to spatial convolution layers with kernel sizes greater than 1, we set its channel compression factor to $\gamma = 4$ to maintain parameter count comparability with other methods. This configuration results in Conv-Adapter having the highest rank among all compared methods' low-rank matrices.

- PiSSA. We employ the official PiSSA implementation from `https://github.com/GraphPKU/PiSSA`. PiSSA is applied to all $1 \times 1$ convolutional layers, as directly reshaping convolution kernels from shape `[C_out, C_in, k, k]` to `[C_out, C_in*k^2]` for standard LoRA application would compromise the network's inherent inductive bias. In practice, we set $\gamma = 20$ to maintain parameter count consistency with other methods. Following established LoRA practices, we use a default scaling factor configuration of $\alpha/r = 2$ as recommended in recent literature.

- HiRA. We adapt the official HiRA implementation from `https://github.com/hqsiswiliam/hira` for convolutional networks. Following the same approach as with PiSSA, we restrict HiRA application exclusively to $1 \times 1$ convolutional layers. To maintain parameter count consistency, we set $\gamma = 20$ and utilize the theoretically derived scaling factors $s_{\text{expect}}$ presented in Section 3.1.

**Training configuration on ADE20K**. For ADE20K segmentation, we employ both a primary decoder head and an auxiliary head. The primary decoder head is optimized using cross-entropy loss (weight = 1.0) and incorporates deep features from all four network stages in both ResNet-50 and ConvNeXt architectures. In contrast, the auxiliary head employs only third-stage features and is trained with a reduced cross-entropy loss weight of 0.4.

For both full fine-tuning and PEFT approaches, we employ the standard 160k iteration schedule. Using a batch size of 16, we perform evaluations at 16k-iteration intervals. In addition to the introduced trainable low-rank matrices, we optimize all bias terms and normalization layers. The PEFT method configurations for ADE20K segmentation remain identical to those used for VTAB-1k classification.

**Training configuration on MS-COCO**. We conduct object detection experiments using the Faster R-CNN framework, adopting the configuration from `https://github.com/SwinTransformer/Swin-Transformer-Object-Detection/tree/master/configs/_base_/models`.
Our implementation differs from Table 11 in three key aspects: (1) a learning rate of $2 \times 10^{-4}$, (2) a standard 500-iteration warmup period, and (3) a batch size of 16 distributed across two NVIDIA A800 GPUs. Following the standard 1x training schedule (`https://github.com/SwinTransformer/Swin-Transformer-Object-Detection/blob/`

Table 12: Average top-1 accuracy on the VTAB-1k NATURAL benchmark across three independent runs. The highest-performing PEFT method for each task is highlighted in **bold**.

| Backbone | Method | #Param | Caltech101 | CIFAR-100 | DTD | Flowers102 | Pets | Sun397 | SVHN | ‖ Average |
|---|---|---|---|---|---|---|---|---|---|---|
| ResNet-50 | FT | 23.5M | 89.49±0.19 | 30.82±0.16 | 64.41±0.67 | 89.35±0.34 | 84.43±0.46 | 31.00±0.17 | 82.51±0.03 | ‖ 67.43±0.29 |
| | Conv-Adapter [32] | 1.4M | 86.98±0.13 | 27.22±0.32 | **64.40**±0.59 | 82.21±0.19 | 88.98±0.08 | 32.67±0.06 | 51.31±0.78 | ‖ 61.97±0.31 |
| | PiSSA [7] | 1.2M | 88.12±0.26 | 26.74±0.49 | 62.93±0.15 | 85.85±0.91 | 89.13±0.05 | **32.71**±0.27 | 63.32±0.30 | ‖ 64.11±0.35 |
| | HiRA [22] | 1.2M | 87.00±0.18 | 27.97±0.07 | 63.99±0.30 | 82.39±0.26 | **89.20**±0.14 | 32.28±0.14 | 53.31±0.64 | ‖ 62.31±0.25 |
| | CoLoRA (ours) | 1.0M | **89.12**±0.24 | **29.98**±0.48 | 62.82±0.74 | **87.51**±0.25 | 88.87±0.45 | 32.38±0.57 | **75.31**±0.78 | ‖ **66.57**±0.50 |
| ConvNeXt-S | FT | 49.5M | 91.33±0.55 | 65.60±0.32 | 74.17±0.05 | 98.74±0.10 | 90.36±0.22 | 49.02±0.19 | 92.30±0.13 | ‖ 80.22±0.22 |
| | Conv-Adapter [32] | 2.8M | 90.94±0.45 | 68.52±0.36 | 75.37±0.40 | 99.12±0.07 | 91.13±0.09 | 49.91±0.06 | 90.35±0.36 | ‖ 80.76±0.26 |
| | PiSSA [7] | 3.0M | 90.93±0.12 | 69.22±0.40 | 75.62±0.10 | 99.06±0.04 | 91.29±0.16 | 51.96±0.29 | 90.27±0.29 | ‖ 81.19±0.20 |
| | HiRA [22] | 3.0M | 90.71±0.24 | **70.85**±0.20 | **76.30**±0.11 | **99.27**±0.00 | **92.12**±0.07 | **53.56**±0.14 | 86.39±0.10 | ‖ **81.31**±0.12 |
| | CoLoRA (ours) | **2.6M** | **91.60**±0.11 | 66.34±0.55 | 75.00±0.60 | 98.89±0.06 | 90.43±0.27 | 49.94±0.33 | **92.08**±0.21 | ‖ 80.61±0.31 |

Table 13: Average top-1 classification accuracy on the VTAB-1k SPECIALIZED benchmark across three experimental trials. Top-performing PEFT methods are indicated in **bold**.

| Backbone | Method | #Param | Camelyon | EuroSAT | Resisc45 | Retinopathy | Average |
|---|---|---|---|---|---|---|---|
| ResNet-50 | FT | 23.5M | 85.31±0.61 | 91.83±0.34 | 80.94±0.11 | 73.82±0.20 | 82.98±0.35 |
| | Conv-Adapter [32] | 1.4M | 78.59±0.28 | 88.20±0.44 | 75.29±0.23 | 73.80±0.06 | 78.97±0.25 |
| | PiSSA [7] | 1.2M | 81.59±0.58 | 90.25±0.55 | 78.47±0.56 | 73.82±0.15 | 81.03±0.46 |
| | HiRA [22] | 1.2M | 79.07±0.10 | 89.10±0.03 | 75.42±0.11 | 73.85±0.02 | 79.36±0.07 |
| | CoLoRA (ours) | 1.0M | **82.60**±0.58 | **92.16**±0.20 | **81.08**±0.26 | **74.30**±0.19 | **82.54**±0.31 |
| ConvNeXt-S | FT | 49.5M | 87.95±0.45 | 95.98±0.06 | 85.98±0.75 | 76.83±0.27 | 86.69±0.38 |
| | Conv-Adapter [32] | 2.8M | 85.75±0.75 | 94.89±0.16 | 84.04±0.24 | 75.30±0.22 | 84.99±0.34 |
| | PiSSA [7] | 3.0M | 85.79±0.07 | 95.45±0.09 | 85.56±0.15 | 75.56±0.26 | 85.59±0.14 |
| | HiRA [22] | 3.0M | 84.29±0.20 | 94.85±0.07 | 84.91±0.06 | 75.85±0.12 | 84.97±0.11 |
| | CoLoRA (ours) | **2.6M** | **87.51**±0.07 | **96.00**±0.12 | **85.95**±0.15 | **77.13**±0.12 | **86.65**±0.11 |

`master/configs/_base_/schedules/schedule_1x.py`), we train models on MS-COCO for 12 epochs with learning rate adjustments at the 8th and 11th epochs. The PEFT method configurations for object detection remain consistent with those used for VTAB-1k classification and ADE20K segmentation.

# I   More Details on VTAB-1k Benchmark

We present comprehensive quantitative results on the VTAB-1k benchmark, including the average top-1 accuracy and its standard deviation in Tables 12 to 14. All methods were evaluated on VTAB-1k across three independent runs. Notably, the proposed CoLoRA significantly outperforms existing PEFT methods in the NATURAL, SPECIALIZED, and STRUCTURED categories with ResNet-50 as the backbone. When using ConvNeXt-S, CoLoRA achieves the highest top-1 accuracy in the SPECIALIZED and STRUCTURED categories, compared to state-of-the-art PEFT methods. Furthermore, CoLoRA surpasses full fine-tuning in the STRUCTURED category, highlighting its superior adaptation capability. Consequently, CoLoRA outperforms full fine-tuning on VTAB-1k with ConvNeXt-S as the backbone.

# J   Scaling CoLoRA on ADE20K

In this section, we investigate the relationship between the number of tunable parameters and adaptation performance. Using the ConvNeXt-B backbone, we evaluate different values for $\gamma$ and the rank ratio $r_s/(r_s + r_l)$. Note that full fine-tuning of ConvNeXt-B on ADE20K achieves 50.99 mIoU. The results, presented in Table 15, demonstrate that CoLoRA with merely 19.5M tunable parameters (compared to 87.6M for full fine-tuning) not only matches but surpasses full fine-tuning performance, reaching 51.13 mIoU. This remarkable performance highlights CoLoRA's potential for efficiently fine-tuning pre-trained ConvNets across diverse downstream tasks.

Table 14: Average top-1 accuracy across three runs on the VTAB-1k STRUCTURED benchmark. The highest-performing PEFT method is highlighted in **bold**.

| Backbone | Method | #Param | Clevr-Count | Clevr-Dist | DMLab | dSpr-Loc | dSpr-Ori | KITTI-Dist | sNORB-Azim | sNORB-Elev | Average |
|---|---|---|---|---|---|---|---|---|---|---|---|
| | FT | 23.5M | 43.27±0.58 | 55.21±0.48 | 45.67±0.14 | 80.18±0.82 | 41.93±0.56 | 80.59±1.11 | 26.54±0.16 | 48.21±1.55 | 52.70±0.67 |
| | Conv-Adapter [32] | 1.4M | 35.94±0.05 | 44.98±1.55 | 35.40±0.36 | 41.50±0.62 | 15.29±0.95 | 69.95±1.04 | 14.72±0.12 | 38.21±0.79 | 37.00±0.67 |
| ResNet-50 | PiSSA [7] | 1.2M | 48.61±0.29 | 49.16±0.63 | 38.21±0.46 | 52.62±1.11 | 22.29±0.32 | 73.70±1.19 | 18.05±0.61 | 39.43±0.89 | 42.76±0.69 |
| | HiRA [22] | 1.2M | 40.98±0.31 | 46.69±0.72 | 35.10±0.40 | 45.70±0.26 | 16.94±0.23 | 71.31±0.75 | 13.53±0.27 | 44.05±0.31 | 39.29±0.41 |
| | CoLoRA (**ours**) | 1.0M | **54.04±3.25** | **53.95±0.86** | **42.05±0.39** | **76.42±1.33** | **37.18±0.53** | **79.98±0.77** | **22.58±0.47** | **48.81±1.23** | **51.88±1.10** |
| | FT | 49.5M | 91.06±0.40 | 66.58±0.60 | 55.10±0.37 | 92.75±0.49 | 62.41±0.77 | 83.88±0.17 | 39.96±0.68 | 46.91±0.18 | 67.33±0.46 |
| | Conv-Adapter [32] | 2.8M | 87.82±0.65 | 66.06±0.62 | 48.68±0.67 | 94.11±0.25 | 61.30±0.47 | **85.04±0.46** | 37.29±0.54 | 49.14±0.45 | 66.18±0.52 |
| ConvNeXt-S | PiSSA [7] | 3.0M | 91.51±0.54 | **67.04±0.90** | 51.35±0.49 | **94.47±0.30** | 61.47±0.65 | 82.79±0.07 | 37.24±0.46 | 49.15±0.05 | 66.88±0.43 |
| | HiRA [22] | 3.0M | 79.64±3.01 | 63.95±0.39 | 47.33±0.48 | 79.60±2.20 | 56.35±0.97 | 82.32±0.77 | 25.85±0.25 | 42.13±0.26 | 59.65±1.04 |
| | CoLoRA (**ours**) | **2.6M** | **91.71±0.18** | 65.57±0.08 | **54.59±0.61** | 92.94±0.52 | **62.28±0.88** | 83.82±0.40 | **40.71±0.67** | **48.95±0.40** | **67.57±0.47** |

Table 15: Quantitative evaluation of CoLoRA scaling effects on ConvNeXt-B for ADE20K adaptation.

| $\gamma$ | 16 | 8 | 8 | 4 | 4 | 4 | 2 | 2 |
|---|---|---|---|---|---|---|---|---|
| $r_s/(r_s + r_l)$ | 0.5 | 0.8 | 0.5 | 0.8 | 0.5 | 0.4 | 0.9 | 0.8 |
| #Params | 4.6M | 5.7M | 8.8M | 11.1M | 17.3M | 19.5M | 17.7M | 21.9M |
| mIoU | 49.55 | 50.65 | 50.86 | 50.86 | 50.96 | **51.13** | 50.38 | 51.06 |

