# OpenReview forum: "Correlated Low-Rank Adaptation for ConvNets"
_NeurIPS.cc/2025/Conference — NeurIPS 2025 poster_

### Official Review · Reviewer_UVug · 2025-06-23

**Clarity:** 3
**Significance:** 2
**Originality:** 3
**Rating:** 4
**Confidence:** 4

**Summary:**

This paper proposes a correlation low-rank adaptation module for convolutional networks, which is an efficient PEFT strategy. The effectiveness of the approach is validated in classification, detection, and segmentation tasks.

**Questions:**

I hope the authors will address the issues with the weaknesses, and I will consider revising my score.

**Ethical Concerns:**

["NO or VERY MINOR ethics concerns only"]

**Final Justification:**

I have carefully read the authors’ responses to my and the other reviewers’ comments, and I have decided to keep my rating at ‘borderline accept.’ Overall, I believe this work falls outside the acceptance criteria for NeurIPS.

**Limitations:**

1. The only shortcoming of this paper is that its performance is slightly lower than that of SFT, which might be a common issue.

**Paper Formatting Concerns:**

Nothing

**Quality:**

3

**Strengths And Weaknesses:**

Strengths:

1. This paper finds that fine-tuning pre-trained ConvNets on downstream tasks is a full-rank learning process, and existing LoRA methods are ineffective within the ConvNets framework.
2. This paper proposes a novel weight update strategy to update adjacent convolution layers together.
3. This paper compares its approach with other PEFT methods on tasks such as object detection, classification, and segmentation, and the results show superior performance.

Weaknesses:
The approach only outperforms SFT in classification tasks, while in detection and segmentation tasks, the performance of CoLoRA is not as good as that of SFT. Without considering the resource consumption of the model, the contribution of CoLoRA would be discounted.

---

> ### Author Rebuttal · Authors · 2025-07-31
>
> We appreciate your positive rating and recognition of the novelty of our method. Below, we respond to your concern regarding the comparative performance of CoLoRA and SFT.
>
> [Q1] **The approach only outperforms SFT in classification tasks, while in detection and segmentation tasks, the performance of CoLoRA is not as good as that of SFT. Without considering the resource consumption of the model, the contribution of CoLoRA would be discounted.** **The only shortcoming of this paper is that its performance is slightly lower than that of SFT, which might be a common issue.**
>
> We previously demonstrated in Table 1 that the proposed CoLoRA method outperforms SFT in classification tasks. In this response, we provide further evidence that CoLoRA either surpasses or matches the performance of SFT in classification, segmentation, and detection tasks.
>
> **Classification on VTAB-1k:** As shown in Table 1, CoLoRA using ConvNeXt-S surpasses SFT in terms of mean top-1 accuracy across the 19 VTAB-1k tasks. This illustrates its effectiveness in transfer learning across diverse visual domains.
>
> **Classification on fine-grained CUB-2011:** As suggested also by other reviewers, we evaluated CoLoRA on CUB-2011, a challenging fine-grained benchmark that includes 200 bird categories. We fine-tuned a pre-trained ConvNeXt-S for 50 epochs with a batch size of 32. The results were as follows: CoLoRA achieved 85.28% top-1 accuracy, while SFT (full-tuning) attained 84.12% top-1 accuracy. This represents a gain of 1.16% over SFT, highlighting CoLoRA's effectiveness in fine-grained tasks.
>
> **Segmentation on ADE20K:** Dense prediction tasks, such as segmentation, present significant challenges for PEFT methods. For instance, the official Conv-Adapter, which has 39.4M tunable parameters on ConvNeXt-B, still falls short by 1.8 mIoU compared to SFT. In contrast, **CoLoRA can match or exceed SFT's performance while utilizing significantly fewer parameters in both configurations:**
>
> - Config 1: with a channel compression factor of $\gamma=4$ and a shared LoRA ratio of 0.4, this configuration results in 19.5M tunable parameters and achieves 51.13 mIoU, slightly outperforming SFT by 0.14 mIoU.
> - Config 2: we set $\gamma=2$ and the shared LoRA ratio to 0.8, which yields 21.9M tunable parameters. This configuration offers a 51.06 mIoU, marginally surpassing the performance of SFT.
>
> These results demonstrate that CoLoRA can **match or exceed full-tuning** performance with **50% fewer trainable parameters**.
>
> *Detection on MS-COCO*. Tuning ConvNets to outperform SFT in detection tasks poses a significant challenge. For example, the Conv-Adapter introduces 24.6M parameters on the ConvNeXt-B model but still falls short of SFT by 3.3 mAP points. To investigate the potential of CoLoRA, we fine-tuned a ConvNeXt-XL model within the Cascade R-CNN framework, utilizing only 8.2M tunable parameters. Remarkably, both CoLoRA and SFT achieved identical performance levels of 52.5 mAP. Despite employing significantly fewer parameters, **CoLoRA matches the performance of full fine-tuning**, demonstrating its effectiveness in complex detection pipelines.
>
> In summary, our extensive experiments across classification (VTAB-1k, CUB-2011), segmentation (ADE20K), and detection (MS-COCO) demonstrate that **CoLoRA can match or exceed SFT** in various tasks, highlighting CoLoRA's practical applicability, scalability, and efficiency for a wide range of vision tasks.

---

> > ### Comment · Reviewer_UVug · 2025-08-01
> >
> > Thank you to the authors for the additional experimental analysis on object detection and segmentation tasks. This paper is valuable, but it is regrettable that the results did not surpass those of the SFT approach. I look forward to more valuable work from the authors in the future.

---

> ### Author Response · Authors · 2025-08-01
> **Response to the reviewer UVug**
>
> Thank you for your response. As demonstrated in our rebuttal, we present **two configurations in which the proposed method *surpasses* SFT on the segmentation task**. For the detection task, it **achieves the identical performance while utilizing only 2.34% of tunable parameters**. In contrast, the current state-of-the-art (SOTA) method continues to fall significantly short of SFT on both segmentation and detection tasks, despite employing over 40% of tunable parameters. We believe these results and comparisons strongly support the effectiveness and potential of our approach across a range of downstream vision tasks.

---

### Official Review · Reviewer_r6JU · 2025-06-30

**Clarity:** 3
**Significance:** 3
**Originality:** 3
**Rating:** 4
**Confidence:** 4

**Summary:**

The paper proposes CoLoRA, a novel Low-Rank Adaptation framework tailored for convolutional networks (ConvNets), addressing the limitations of standard LoRA in highly correlated convolutional layers. Unlike existing LoRA methods that treat layers independently, CoLoRA introduces correlated low-rank matrices to model inter-layer dependencies. To further enhance adaptation efficiency, it incorporates a parameter-free filtering mechanism that expands the receptive field and reduces noise from uninformative regions. Extensive experiments across classification, segmentation, and detection show that CoLoRA consistently outperforms prior PEFT methods. Remarkably, it achieves better performance than full fine-tuning on VTAB-1k using only 5% of trainable parameters.

**Questions:**

Please refer to the previously mentioned weaknesses.

Additionally, I would like to understand the degree of correlation between adjacent attention layers in Vision Transformers (ViTs). Is this correlation significantly different from that observed in convolutional networks, thereby making CoLoRA specifically tailored for convolutional architectures?

**Ethical Concerns:**

["NO or VERY MINOR ethics concerns only"]

**Final Justification:**

The authors have addressed most of my concerns through their detailed rebuttal, and I decide to keep my rating as "Borderline Accept". Overall, I think this work is above the bar of the acceptance of NeurIPS.

**Limitations:**

Yes.

**Paper Formatting Concerns:**

No.

**Quality:**

3

**Strengths And Weaknesses:**

Strengths:
1. The proposed Correlated Low-Rank Adaptation (CoLoRA) is both novel and well-motivated, supported by the empirical observation of inter-layer correlations in convolutional networks, as illustrated in Figure 2.
2. The paper provides a clear and insightful analysis of the spatial limitations of LoRA in ConvNets, and the introduced training-free edge-preserving filtering mechanism (Figure 4) offers a compelling and practical solution.
3. CoLoRA demonstrates notable performance improvements over state-of-the-art PEFT methods on multiple vision benchmarks, including VTAB-1K, COCO, and ADE20K, with a comparable number of trainable parameters.
4. The ablation studies presented in Table 3 are thorough and well-designed, effectively validating the contribution of each component within the proposed framework.


Weaknesses:
1. Unlike standard LoRA, CoLoRA applies edge-preserving filtering to the compressed features generated by low-rank projections, which appears to preclude re-parameterization into static weights. As a result, inference-time efficiency may be impacted due to the additional computation. It would be important to quantify this overhead by reporting inference time and computational cost comparisons between CoLoRA and standard LoRA across both classification and detection tasks.
2. While CoLoRA is evaluated on VTAB-1K, COCO, and ADE20K, many recent visual PEFT methods also benchmark on fine-grained visual classification (FGVC) datasets. A comparison with state-of-the-art methods in this domain would strengthen the generality and competitiveness of the proposed approach.

---

> ### Author Rebuttal · Authors · 2025-07-31
>
> We thank the reviewer for the constructive comments and the positive score, especially the recognition of our method as both novel and well-motivated. We address the raised concern in detail below.
>
> [Q1] **Unlike standard LoRA, CoLoRA applies edge-preserving filtering to the compressed features generated by low-rank projections, which appears to preclude re-parameterization into static weights. As a result, inference-time efficiency may be impacted due to the additional computation. It would be important to quantify this overhead by reporting inference time and computational cost comparisons between CoLoRA and standard LoRA across both classification and detection tasks.**
>
> We acknowledge the reviewer’s concern regarding the inference-time efficiency of CoLoRA. In our design, the LoRA weights can be merged into the pre-trained weights, similar to PiSSA and HIRA. Therefore, **the additional computational cost arises solely from the edge-preserving filtering**.
>
> To address this overhead, we have implemented several optimizations. These include the use of `torch.jit.script`to accelerate the filtering process. We now provide a detailed comparison of inference time across three representative tasks: classification, detection, and segmentation. All experiments use the ConvNeXt-S backbone.
>
> *Classification task*. We evaluate the inference time with ConvNeXt-S backbone on FGVC aircraft dataset, where image resolution is 224x224.
>
> | Method                                | Inference Time (s) |
> | ------------------------------------- | ------------------ |
> | PiSSA, HIRA, and CoLoRA w/o Filtering | 0.042              |
> | CoLoRA                                | 0.053              |
>
> *Detection task*. We evaluate the inference time under the Faster-RCNN framework with ConvNeXt-S backbone. The results are tabulated below:
>
> | Method                                | Inference Time (s) |
> | ------------------------------------- | ------------------ |
> | PiSSA, HIRA, and CoLoRA w/o Filtering | 0.108              |
> | CoLoRA                                | 0.108              |
>
> *Segmentation task*. We evaluate the inference time with ConvNeXt-S backbone on ADE20K dataset. The results are shown below:
>
> | Method                                | Inference Time (s) |
> | ------------------------------------- | ------------------ |
> | PiSSA, HIRA, and CoLoRA w/o Filtering | 0.250              |
> | CoLoRA                                | 0.260              |
>
> **In summary, for classification tasks, where most of the computational cost is concentrated in the backbone, the filtering module introduces a moderate increase in inference time. In contrast, for detection and segmentation tasks, the additional overhead caused by filtering is comparatively negligible.**
>
> [Q2] **While CoLoRA is evaluated on VTAB-1K, COCO, and ADE20K, many recent visual PEFT methods also benchmark on fine-grained visual classification (FGVC) datasets. A comparison with state-of-the-art methods in this domain would strengthen the generality and competitiveness of the proposed approach.**
>
> We appreciate this valuable suggestion. To further assess the generality of our method, we conduct experiments on two widely-used FGVC datasets: CUB-2011 and FGVC Aircraft. The CUB-2011 dataset contains 200 bird species, comprising 5,994 training and 5,794 testing images. For the FGVC Aircraft dataset, we adopt the 'variant' split, which includes 6,667 training and 3,333 testing images spanning 102 fine-grained categories.
>
> We fine-tune a pre-trained ConvNeXt-S backbone on both datasets for 50 epochs using a batch size of 32. The top-1 accuracy results are summarized below:
>
> |               | Full-tuning | PiSSA+merging | HIRA+merging | CoLoRA |
> | :-----------: | :---------: | :-----------: | :----------: | :----: |
> |   CUB-2011    |    84.12    |     84.90     |    83.47     | 85.28  |
> | FGVC aircraft |    83.65    |     79.42     |    53.20     | 80.71  |
>
> CoLoRA consistently outperforms both PiSSA and HIRA across these FGVC benchmarks. On CUB-2011, it achieves a 1.16% improvement over full fine-tuning and a 0.38% gain over PiSSA. On the FGVC Aircraft dataset, CoLoRA surpasses PiSSA by 1.29%. These results further confirm the effectiveness and competitiveness of our method in fine-grained classification tasks. We will include these findings in the revised manuscript to strengthen the generality of our contributions.
>
> [Q3] **Additionally, I would like to understand the degree of correlation between adjacent attention layers in Vision Transformers (ViTs). Is this correlation significantly different from that observed in convolutional networks, thereby making CoLoRA specifically tailored for convolutional architectures?**
>
> We thank the reviewer for this insightful question. In our current work, CoLoRA is primarily designed for convolutional networks, where the strong correlation between adjacent convolution layers motivates our design. However, exploring the applicability of CoLoRA to Vision Transformers (ViTs) is indeed a valuable direction.
>
> To investigate this, we fine-tune a pre-trained ViT-B model on two FGVC datasets and analyze the degree of correlation between adjacent modules. In particular, we examine: (1) the correlation between the QKV projection and the output projection in the self-attention block, and (2) the correlation between the two linear layers in the feed-forward network (FFN). We use the **Centered Kernel Alignment (CKA)** metric to measure correlation, as it is invariant to orthogonal transformation and well-suited for comparing representational similarity in ViTs. Our findings suggest that the **correlation between adjacent layers in ViTs is substantially weaker than that observed in ConvNets**. Despite this, we evaluate CoLoRA on ViTs by fine-tuning ViT-B on CUB-2011 and FGVC Aircraft datasets. The results are shown below:
>
> |               | Full-tuning | PiSSA | HIRA  | CoLoRA |
> | :-----------: | :---------: | :---: | :---: | :----: |
> |   CUB-2011    |    79.24    | 74.70 | 51.97 | 75.61  |
> | FGVC aircraft |    79.30    | 72.40 | 34.32 | 72.79  |
>
> Although the inter-layer correlation in ViTs is weaker, CoLoRA still outperforms existing PEFT methods such as PiSSA and HIRA on both datasets. However, its performance does not match that of full fine-tuning, suggesting that **the adaptation of CoLoRA to ViTs requires further investigation and architectural refinement**. We consider this an important limitation and will clarify it in the revised manuscript.

---

> > ### Comment · Reviewer_r6JU · 2025-08-06
> >
> > The authors have addressed most of my concerns through their detailed rebuttal, and I decide to keep my rating as "Borderline Accept". Overall, I think this work is above the bar of the acceptance of NeurIPS.

---

> > > ### Author Response · Authors · 2025-08-07
> > > **Response from the Authors**
> > >
> > > Thank you very much for your thoughtful review and for acknowledging that our rebuttal has addressed most of your concerns. We sincerely appreciate your assessment that our work is above the bar for acceptance at NeurIPS.

---

### Official Review · Reviewer_fmfm · 2025-07-02

**Clarity:** 3
**Significance:** 2
**Originality:** 3
**Rating:** 4
**Confidence:** 4

**Summary:**

This paper proposes a novel method to fine-tune convolutional neural network taking into account the correlation between adjacent convolutional layers. Authors decompose low-rank matrix into a sum of two low-rank matrices: one of them is shared across the pair of adjacent layers and the second one is layer-specific. Apart from that, authors introduce edge-preserving convolutional filter consisting of 4 convolutions to expand the receptive field of their fine-tuning method.

**Questions:**

1. Did you try any other similarity metrics (for example Frobenius scalar product or distance between Images) to support the idea of the adjacent layers’ correlation?
2. How much does applying edge-preserving filter on every step affect fine-tuning time?
3. Do you iteratively merge low-rank adapters for all methods in your experiments? It seems that there is also no comparison to vanilla LoRA or its version with the iterative merging. Therefore, it is difficult to identify the effect solely from weight sharing.

**Ethical Concerns:**

["NO or VERY MINOR ethics concerns only"]

**Final Justification:**

The results became more convincing after the rebuttal. Among the drawbacks, there are currently no error bars.

**Limitations:**

Yes

**Paper Formatting Concerns:**

-

**Quality:**

2

**Strengths And Weaknesses:**

Strengths:
* Authors consecutively apply their theoretical findings one by one in their experiments showing a gradual increase in quality metrics, supporting usability of each upgrade.
* This approach introduces an idea that we can exploit the correlation between matrices of adjacent layers and use a shared matrix in their adapters. This idea is quite interesting and could inspire future research.

Weaknesses:
* The authors argue that ConvNext-B requires a full-rank matrix update via checking the matrix rank of the update after full fine-tuning. However, this does not guarantee that full-rank updates are the only viable option. Imposing structural constraints during training may lead to a different set of parameters. A more rigorous approach would be to evaluate the dependence on rank by fine-tuning the model multiple times with varying ranks and comparing performance metrics across different parameter budgets.
* The choice of similarity metric for weight update matrices seems unjustified. The proximity of mean singular values for adjacent layers, shown in Figure 2, shows only the relative poximity of Frobenius norms of weight updates, and may not imply real correlation between weight updates. Furthermore, the mean singular value of the weight update primarily depends on the matrix entries magnitude, which, in particular, depends on the learning rate schedule.
* Transition from (3) and (4) to (5) requires more details and explanations.

---

> ### Author Rebuttal · Authors · 2025-07-31
>
> We sincerely thank the reviewer for the high-quality review and insightful comments. We are also delighted that the reviewer finds our proposed method interesting and potentially inspiring for future research. Below, we address the raised concerns in detail.
>
> [Q1] **The authors argue that ConvNext-B requires a full-rank matrix update via checking the matrix rank of the update after full fine-tuning. However, this does not guarantee that full-rank updates are the only viable option. Imposing structural constraints during training may lead to a different set of parameters. A more rigorous approach would be to evaluate the dependence on rank by fine-tuning the model multiple times with varying ranks and comparing performance metrics across different parameter budgets.**
>
> We appreciate your insightful observation. We agree that inspecting the rank of update weights alone is insufficient to conclude that ConvNeXt-B inherently requires full-rank updates. To provide a more rigorous justification, we conducted experiments by tuning ConvNeXt-B using PiSSA under varying ranks.
>
> For ConvNets, we parameterize the rank via a channel compression factor $\gamma$, where  $r=d/\gamma$ and $d$ is the number of channels. Notably, $\gamma=1$ corresponds to full-rank adaptation. It's worth mentioning that LoRA introduces an over-parameterization when $\gamma=1$ due to the decomposition $W=sBA$, resulting in more trainable parameters than the original full-rank $W$. Quantitative results are presented below:
>
> | $\gamma$            | 1      | 2     | 4     | 8     | 20   |
> | ------------------- | ------ | ----- | ----- | ----- | ---- |
> | tunable params      | 104.8M | 52.5M | 26.3M | 13.2M | 5.3M |
> | tunable param ratio | 120%   | 60%   | 30%   | 15%   | 6%   |
> | mIoU                | 50.1   | 50.6  | 49.9  | 49.1  | 47.6 |
>
> As shown above, performance peaks at $\gamma=2$, while both under- and over-parameterization degrade performance. Increasing $\gamma$ leads to a consistent drop in mIoU from 50.6 to 47.6. This validates that **independent LoRA tuning on ConvNeXt-B requires high-rank adaptation**, and performance deteriorates under tight parameter budgets. We will include this analysis in the revised manuscript to reinforce our argument.
>
> [Q2] **The choice of similarity metric for weight update matrices seems unjustified. The proximity of mean singular values for adjacent layers, shown in Figure 2, shows only the relative poximity of Frobenius norms of weight updates, and may not imply real correlation between weight updates. Furthermore, the mean singular value of the weight update primarily depends on the matrix entries magnitude, which, in particular, depends on the learning rate schedule.** **Did you try any other similarity metrics (for example Frobenius scalar product or distance between Images) to support the idea of the adjacent layers’ correlation?**
>
> We gratefully appreciate the insightful comment. In the paper, we discussed the correlation between weights of adjacent layers (e.g., $W_1$ and $W_2$) based on shared gradient terms. Specifically, during backpropagation, the term $X^T(\partial L/\partial Y_2)$ appears in both gradients $\Delta W_1$ and $\Delta W_2$, suggesting inherent correlation.
>
> In the manuscript, we measured the mean singular value of these update matrices to understand their behavior. However, we agree that this measure is influenced by the magnitude of entries and the learning rate schedule, and does not directly reflect structural correlation.
>
> Following your suggestion, we applied the Forbenius scalar product as well as the Centered Kernel Alignment (CKA) metric to explore the correlation between adjacent layers. CKA is invariant to isotropic scaling, orthogonal transformations, and invertible linear mappings, making it a robust metric for capturing correlation between different layers. This provides a more principled analysis of inter-layer correlation, and we will include these results in the revised version.
>
> [Q3] **Transition from (3) and (4) to (5) requires more details and explanations.**
>
> We gratefully appreciate the constructive suggestion.
>
> Starting from Eqs. (3) and (4):
>
> $$\mathbf{g}_{W_2}=W_1^TX^T(\partial L/\partial Y_2) + [\mathbf{b}_1,\cdots, \mathbf{b}_1]_N (\partial L/\partial Y_2),$$
>
> $$\mathbf{g}_{W_1}=X^T(\partial L/\partial Y_2)W^T_2 + \mathbf{0}.$$
>
> Let us denote the shared term $\Lambda = X^T(\partial L/\partial Y_2)$, and the bias-related term for $\mathbf{g}_{W_2}$ as $\Psi$. The gradients can then be rewritten as
>
> $$\mathbf{g}_{W_2}=W_1^T \Lambda + \Psi$$
>
> $$ \mathbf{g}_{W_1} = \Lambda W_2^T$$
>
> Under gradient descent with learning rate $\eta$ (which is widely employed to understand the behavior of LoRA), we have:
>
> $$\Delta W_1=SW^T_2 + P_1,\ \Delta W_2=W_1^T S + P_2,$$
>
> where $S=-\eta \Lambda$, $P_2=-\eta \Psi$, and $P_1=\mathbf{0}$. To integrate this behavior to LoRA, we decompose $S$, $P_1$, $P_2$ into learnable low-rank matrices, forming the basis of our proposed CoLoRA.
>
> [Q4] **How much does applying edge-preserving filter on every step affect fine-tuning time?**
>
> We thank the reviewer for this important and practical question. Below, we provide a comprehensive comparison of GPU memory usage, per-step training time, and inference time across different tuning strategies.
>
> At inference time, the LoRA weights of PiSSA, HIRA, and CoLoRA are merged into the base model weights. Therefore, the inference time for full-tuning, PiSSA, and HIRA remains identical. The additional inference time of CoLoRA arises solely from the edge-preserving filtering module. To minimize this overhead, we adopt efficient implementations based on `torch.jit.script`.
>
> During training, all LoRA-based methods (PiSSA, HIRA, and CoLoRA) introduce additional memory and computation due to the intermediate low-rank matrices. For CoLoRA, the filtering operation and the separate computation of shared and independent LoRA branches lead to further computational overhead. Moreover, **mixed-precision training using APEX introduces memory usage variations depending on bits of parameters**. We provide details below:
>
> *Computation comparison on detection*. We evaluate the time efficiency with ConvNeXt-S backbone on the MS-COCO dataset, where the batch size is 2 with `coco_instance.py` dataset configuration.
>
> |                                            | Full-tuning | PiSSA | HIRA  | CoLoRA | CoLoRA w/o filtering |
> | ------------------------------------------ | ----------- | ----- | ----- | ------ | -------------------- |
> | **Training GPU memory consummation (MiB)** | 18418       | 14648 | 14536 | 15042  | 16488                |
> | **Per step training time (s)**             | 1.747       | 1.426 | 1.501 | 1.527  | 1.509                |
> | **Inference time (s)**                     | 0.108       | 0.108 | 0.108 | 0.108  | 0.108                |
>
> *Computation comparison on classification*. We evaluate the time efficiency with ConvNeXt-S backbone on the FGVC aircraft dataset, where the batch size is 32 and the image resolution is 224x224.
>
> |                                            | Full-tuning | PiSSA  | HIRA   | CoLoRA | CoLoRA w/o filtering |
> | ------------------------------------------ | ----------- | ------ | ------ | ------ | -------------------- |
> | **Training GPU memory consummation (MiB)** | 7247        | 6277   | 6189   | 7243   | 7097                 |
> | **Per batch training time (s)**            | 0.324      | 0.339 | 0.341 | 0.414  | 0.409                |
> | **inference time (s)**                     | 0.042       | 0.042  | 0.042  | 0.053  | 0.042                |
>
> In summary, our optimized implementation ensures that edge-preserving filtering is not a major bottleneck during training. The slight increase in training time for CoLoRA mainly results from the additional computation of shared and independent LoRA branches that need optimization. **The filtering overhead during inference is minimal for detection task, and moderate for classification**.
>
> [Q5] **Do you iteratively merge low-rank adapters for all methods in your experiments? It seems that there is also no comparison to vanilla LoRA or its version with iterative merging. Therefore, it is difficult to identify the effect solely from weight sharing.**
>
> We appreciate this observation. In our experiments:
>
> - We do not apply LoRA merging to PiSSA or HiRA by default.
> - HiRA already performs high-rank updates and shows little benefit from merging. In fact, merging often introduces oscillations in fine-grained classification tasks.
> - PiSSA uses SVD-based initialization, making LoRA merging non-trivial. It requires repeated SVD and careful reinitialization of $A$ and $B$. Short-interval LoRA merging may introduce noise into SVD decomposition results, while long-interval LoRA merging diminishes performance.
>
> Additionally, to ensure a fair comparison in terms of LoRA merging, we conduct experiments on two FGVC benchmark datasets, CUB-2011 and FGVC aircraft, as suggested by other reviewers. We fine-tune a pre-trained ConvNeXt-S model on these datasets with a batch size of 32 for 50 epochs. Specifically, we merge LoRA weights for PiSSA, HIRA, and CoLoRA every 500 steps, followed by a careful optimizer reset and 50 warmup steps. Quantitative results are presented below:
>
> |               | Full-tuning | PiSSA+merging | HIRA+merging | CoLoRA |
> | :-----------: | :---------: | :-----------: | :----------: | :----: |
> |   CUB-2011    |    84.12    |     84.90     |    83.47     | 85.28  |
> | FGVC aircraft |    79.24    |     74.70     |    51.97     | 75.61  |
>
> These results confirm that **weight sharing, not merging, is the key driver** of CoLoRA’s superior performance.
>
> Finally, for object detection, we found that merging LoRA weights degrades performance, likely due to the multi-branch prediction and the sensitivity of detection heads. Therefore, we disable merging for detection tasks. We will add these results and clarifications to the revised manuscript.

---

> > ### Comment · Reviewer_fmfm · 2025-08-05
> >
> > Thank you for the detailed rebuttal. Most of my concerns have been addressed. Overall, I find that the idea is interesting, and with the new experiments, it is now more justified in my opinion. Therefore, I will raise my score. On the flip side, there are still some technical concerns that I believe may be valuable to address in the revised version of the paper. For example, the result can be sensitive to the choice of the fine-tuning initialization (such as PiSSA, and moreover, there is no pure LoRA in some of your settings). Also, even within one method, there may be large variations of the metrics during different runs.
> >
> > Also, out of curiosity, combining layers in pairs is intuitive, but did you think about more than two years? It seems that your logic with the gradient can be extended to these cases as well.

---

> > > ### Author Response · Authors · 2025-08-06
> > > **Official Response from Authors**
> > >
> > > Thank you very much for your updated rating and for recognizing the strengths of our work. We are pleased that the additional experiments have helped to clarify and justify our approach, while addressing most of your earlier concerns.
> > >
> > > We sincerely appreciate your continued technical suggestions and will incorporate them carefully into the revised manuscript. In particular, we will provide a more detailed discussion on the sensitivity to LoRA initialization and include results using a pure LoRA baseline. To mitigate potential performance variation, we will also report evaluations over multiple independent runs, along with their statistics—*in fact, such statistics from multiple runs have already been included for the VTAB-1k benchmark*.
> > >
> > > Moreover, your suggestion of exploring combinations beyond two layers is insightful. *We have already considered a simple extension in which each layer is correlated with both its preceding and succeeding layers, enabling convolutional layers to be linked in a chain with only a modest increase in computation*. We plan to explore this direction further as part of our future work.

---

### Official Review · Reviewer_wEks · 2025-07-05

**Clarity:** 3
**Significance:** 3
**Originality:** 3
**Rating:** 4
**Confidence:** 4

**Summary:**

The CoLoRA framework proposed in the paper solves the high-rank learning bottleneck in ConvNets fine-tuning and significantly improves the performance of the PEFT method. The paper analyzes the gradient correlation between ConvNets layers in detail and verifies the necessity of correlation modeling through mathematical formulas and experiments.

**Questions:**

Question：
1.	There are typos in the text. It is recommended to check and modify it.
2.	Although some limitations are mentioned in the appendix, the main text does not clearly point out the possible limitations of the method. For example:
3.	Is CoLoRA applicable to deeper networks (such as ConvNeXt-XL) or more complex tasks (such as video processing).
4.	Is the method sensitive to the initialization of low-rank matrices, and its performance under different data distributions.

**Ethical Concerns:**

["NO or VERY MINOR ethics concerns only"]

**Final Justification:**

Given that part of my concerns are addressed, I decide to keep the socre.

**Limitations:**

Yes

**Quality:**

3

**Strengths And Weaknesses:**

Strengths：
This work shows that CoLoRA outperforms existing methods in multiple mainstream visual tasks (classification, segmentation, and detection). The experimental settings are reasonable, based on public datasets and models, and highly reproducible. Using only about 5% of the trainable parameters, the performance can reach or even exceed that of full fine-tuning, demonstrating the practical application potential of the method.


Weaknesses：
The paper contains typos that need correction, and although some limitations are discussed in the appendix, the main text does not clearly address potential shortcomings. Considering there are lots of works on efficient finetuning, the proposed method might be somewhat incremental.

Overall, its strengths has higher weight than its weakness.

---

> ### Author Rebuttal · Authors · 2025-07-31
>
> We sincerely thank the reviewer for the positive score and the recognition of the potential of our proposed method. Below, we carefully address the raised concerns.
>
> [Q1] **There are typos in the text.**
>
> We gratefully appreciate your careful review and pointing out the typos in our manuscript. We have thoroughly proofread and corrected these issues. For example:
>
> - "ConveNets" on page 1 has been corrected to "ConvNets".
> - "Correlated Low RAnk (CoLoRA)" on page 2 has been revised to "Correlated Low-Rank Adaptation (CoLoRA)".
>
> [Q2] **Although some limitations are mentioned in the appendix, the main text does not clearly point out the possible limitations of the method. For example: 3. Is CoLoRA applicable to deeper networks (such as ConvNeXt-XL) or more complex tasks (such as video processing). 4. Is the method sensitive to the initialization of low-rank matrices, and its performance under different data distributions.**
>
> *Limitations*: We acknowledge that, while we briefly mentioned the “correlation strength issue”, the profound limitations were not clearly articulated in the main paper. The **primary limitation** of our proposed CoLoRA method lies in its **computational overhead**. Specifically, the introduction of both shared and layer-specific LoRA modules, as well as the proposed filtering technique, results in increased computational load compared to recent LoRA methods such as PiSSA and HIRA on classification task.
>
> Additionally, we conducted new experiments using ViT backbones on FGVC datasets. While CoLoRA significantly outperforms PiSSA and HiRA in this setting, it still lags behind full fine-tuning by over 3 top-1 accuracy points, suggesting that adapting CoLoRA to ViTs requires further architectural exploration. **These limitations will be incorporated into the revised manuscript and we will move it to the main text**.
>
> *Applicability to Deeper Networks or Complex Tasks*: We have evaluated CoLoRA on deeper architectures, including **ConvNeXt-XL** and **Vision Transformers (ViT)**, to assess scalability and generalization.
>
> - For ConvNeXt-XL, we conducted semantic segmentation on ADE20K for 64k iterations. The results are:
>
> |      | Full-tuning | PiSSA | HIRA  | CoLoRA |
> | :--: | :---------: | :---: | :---: | :----: |
> | mIoU |    53.21    | 51.77 | 50.61 | 52.41  |
>
> Please notice that CoLoRA demonstrates superior performance compared to PiSSA and HIRA, as indicated by its higher mIoU metric, reflecting better convergence. Additionally, we fine-tuned ConvNeXt-XL on the MS-COCO detection dataset using the CascadeRCNN framework. In this process, we found that CoLoRA achieves performance comparable to full tuning while utilizing only 2.34% of the tunable parameters after 110,000 iterations, resulting in an impressive mAP of 52.5.
>
> - For ViTs, we tested on CUB-2011 (200 bird species, 5994 train / 5794 test) and FGVC-Aircraft (variant split) (6667 train / 3333 test, 102 categories). We fine-tuned a ViT-B model for 50 epochs using a batch size of 32. Quantitative results are reported below:
>
> |               | Full-tuning | PiSSA | HIRA  | CoLoRA |
> | :-----------: | :---------: | :---: | :---: | :----: |
> |   CUB-2011    |    79.24    | 74.70 | 51.97 | 75.61  |
> | FGVC aircraft |    79.30    | 72.40 | 34.32 | 72.79  |
>
> While CoLoRA was initially designed for ConvNets, it still shows better performance than recent PEFT baselines on ViTs. However, the distance to full fine-tuning indicates that **applying CoLoRA to ViTs is still an open area for exploration** and a recognized limitation.
>
> *Sensitivity to Initialization and Data Distribution*. Our experiments show that CoLoRA is robust to various initialization schemes and shifts in data distribution. In the VTAB-1k benchmark, we found that CoLoRA achieves comparable or lower standard deviations in top-1 accuracy compared to other methods, particularly on the SPECIALIZED and STRUCTURED subsets, which are known to experience significant distribution shifts.
>
> In summary, we evaluated CoLoRA across a diverse range of datasets, including ADE20K, MS-COCO, VTAB-1k, plate_number_1, and FGVC-Aircraft, demonstrating strong performance in various settings.

---

> > ### Comment · Reviewer_wEks · 2025-08-05
> > **Part of my concerns are addressed, I decide to keep the socre**
> >
> > Given that part of my concerns are addressed, I decide to keep the socre.

---

> > > ### Author Response · Authors · 2025-08-06
> > > **Response from the Authors**
> > >
> > > Thank you for taking the time to review our rebuttal and for acknowledging that we have addressed some of your concerns. We truly value your feedback and will take it into account in the revised manuscript to further strengthen our work.
> > >
> > > *If there are specific aspects that require additional discussion or clarification to address your remaining concerns, we would be glad to elaborate*.

---

### Author Response · Authors · 2025-08-06
**Summary of the Rebuttal**

**Summary of the Rebuttal**

**Comments:** We sincerely appreciate the insightful feedback and constructive suggestions provided by all reviewers. In response, we have conducted additional experiments and provided detailed clarifications to further demonstrate the superiority of the proposed method and to address the raised concerns. Below, we summarize the key findings from these new experiments.

Experiments:

- We performed extensive experiments to validate the **high-rank tuning property of ConvNeXt-B in the segmentation task by evaluating the model under different rank configurations**.

- We extended the proposed CoLoRA framework to Vision Transformers (ViTs) and evaluated it on two FGVC benchmarks, CUB-2011 and FGVC Aircraft. **Results show that our method consistently outperforms state-of-the-art approaches, including PiSSA and HIRA.**
- On the same FGVC benchmarks, we further evaluated **ConvNeXt-S**. **Our method surpasses full fine-tuning by 1.16% in top-1 accuracy**, while maintaining parameter efficiency.
- We conducted a **comprehensive computational analysis** comparing different methods, including GPU memory usage, training time per step, and inference time.
- We show that the proposed method can **surpass SFT on image classification and segmentation, and achieve the identical performance on detection task**. However, current SOTA methods still lags behind the SFT.

We believe that these additional results and clarifications further strengthen the contribution and validity of our submission, and we hope they address the reviewers’ concerns satisfactorily.

---

### Note · Authors · 2025-08-15

We sincerely thank the AC and all reviewers for their constructive feedback and active participation throughout the review and rebuttal process.

We are encouraged that our work received positive overall scores, with reviewers recognizing both its novelty and strong potential for practical applications. In response to reviewer suggestions, we conducted additional experiments covering:

- **Vision Transformer (ViT) architectures. our method surpasses all current SOTA methods even in this challenging setting**
- **Fine-Grained Visual Classification (FGVC) on CUB-2011. Our approach outperforms SFT, demonstrating strong adaptability to fine-grained tasks**
- **Segmentation and detection tasks. Our method outperforms SFT in segmentation and matches SFT in detection, while exceeding the performance of other SOTA methods**
- **The proposed method stands out among current approaches by surpassing or matching SFT on classification, segmentation, and detection tasks when applied to ConvNets.**

These extended results, alongside our clarifications, stimulated constructive discussion during the rebuttal period. Notably:
- **We addressed most concerns raised by reviewers fmfm and r6JU, and partially addressed those of wEks. And we believe that our additional results could address the concern of the reviewer UVug**
- **Three reviewers maintained their positive scores, and one reviewer indicated he/she would raise the score to positive**
- **One reviewer explicitly stated: “I think this work is above the bar of the acceptance of NeurIPS"**

In summary, the reviewers have characterized our proposed CoLoRA as novel and interesting. With extensive experimental validation across classification, segmentation, and detection tasks, we believe this work will **inspire future research in parameter-efficient fine-tuning for convolutional architectures and provide tangible benefits to the broader community**.

---

### Decision · Program_Chairs · 2025-09-17

**Decision:**

Accept (poster)

**Comment:**

This paper proposes CoLoRA, a novel parameter-efficient fine-tuning method designed for convolutional networks that explicitly models inter-layer correlations through shared and layer-specific low-rank matrices, complemented by a parameter-free edge-preserving filtering module. The method demonstrates strong empirical performance, outperforming state-of-the-art PEFT baselines across classification, segmentation, and detection tasks, and even surpassing full fine-tuning on VTAB-1k and fine-grained benchmarks while requiring only 5% of trainable parameters. Reviewers acknowledged the novelty of correlation modeling for ConvNets, the thorough experimental validation, and the potential of this work to inspire future research in efficient adaptation for non-transformer architectures. Concerns were raised about being incremental novelty, computational overhead compared to other LoRA variants, and performance gaps when extending to Vision Transformers, although these were largely addressed through additional experiments and clarifications in the rebuttal. Three reviewers indicated positive assessments. Overall, the strengths in novelty, reproducibility, and breadth of validation outweigh the weaknesses, and as such I recommend the paper for acceptance.